# Add and Thin: Diffusion for Temporal Point Processes

David Lüdke[1,2]    Marin Biloš[1,3]    Oleksandr Shchur[1,4,†]
Marten Lienen[1,2]    Stephan Günnemann[1,2]

[1]School of Computation, Information and Technology, Technical University of Munich, Germany
[2]Munich Data Science Institute, Technical University of Munich, Germany
[3]Machine Learning Research, Morgan Stanley, United States
[4]Amazon Web Services, Germany
{d.luedke,m.bilos,o.shchur,m.lienen,s.guennemann}@tum.de

## Abstract

Autoregressive neural networks within the temporal point process (TPP) framework have become the standard for modeling continuous-time event data. Even though these models can expressively capture event sequences in a one-step-ahead fashion, they are inherently limited for long-term forecasting applications due to the accumulation of errors caused by their sequential nature. To overcome these limitations, we derive ADD-THIN, a principled probabilistic denoising diffusion model for TPPs that operates on entire event sequences. Unlike existing diffusion approaches, ADD-THIN naturally handles data with discrete and continuous components. In experiments on synthetic and real-world datasets, our model matches the state-of-the-art TPP models in density estimation and strongly outperforms them in forecasting.

## 1 Introduction

Many machine learning applications involve the analysis of continuous-time data, where the number of events and their times are random variables. This data type arises in various domains, including healthcare, neuroscience, finance, social media, and seismology. Temporal point processes (TPPs) provide a sound mathematical framework to model such event sequences, where the main problem is finding a parametrization that can capture the seasonality and complex interactions (e.g., excitation and inhibition) within point processes.

Traditional TPP models [17, 20] employ simple parametric forms, limiting their flexibility to capture the intricacies of arbitrary TPPs. In recent years, various neural TPPs have been proposed (see [43] for an overview) that capture complex event interactions in an autoregressive manner,

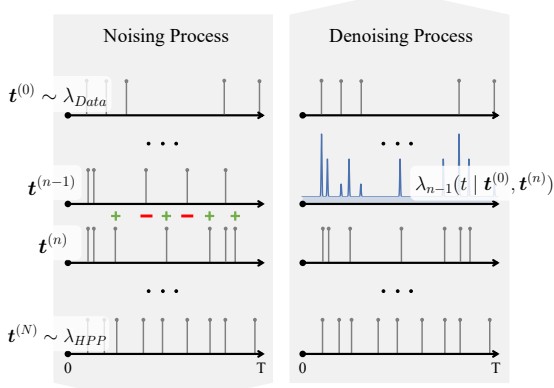

Figure 1: Proposed noising and denoising process for ADD-THIN. *(Left)* Going from step $n-1$ to step $n$, we add and remove some points at random. *(Right)* Given $\boldsymbol{t}^{(n)}$ and $\boldsymbol{t}^{(0)}$ we know the intensity of points at step $n-1$. We approximate this intensity with our model, which enables sampling new sequences.

---

Code is available at https://www.cs.cit.tum.de/daml/add-thin

[†]Work done while at the Technical University of Munich.

37th Conference on Neural Information Processing Systems (NeurIPS 2023).

often using recurrent neural networks (RNN) or transformer architectures. While autoregressive models are expressive and have shown good performance for *one-step-ahead* prediction, their suitability for forecasting remains limited due to the accumulation of errors in sequential sampling.

We propose to take a completely different approach: instead of autoregressive modeling, we apply a generative diffusion model, which iteratively refines entire event sequences from noise. Diffusion models [18, 45] have recently shown impressive results for different data domains, such as images [15, 18, 25], point clouds [31, 32], text [27] and time-series [1, 3, 24, 46]. But how can we apply diffusion models to TPPs? We cannot straightforwardly apply existing Gaussian diffusion models to learn the mapping between TPPs due to the particular requirements that must be met, namely, the randomness in the number of events and the strictly positive arrival times.

We present a novel diffusion-inspired model for TPPs that allows sampling entire event sequences at once without relying on a specific choice of a parametric distribution. Instead, our model learns the probabilistic mapping from complete noise, i.e., a homogeneous Poisson process (HPP), to data. More specifically, we learn a model to reverse our noising process of adding (superposition) and removing (thinning) points from the event sequence by matching the conditional inhomogeneous denoising intensity $\lambda_{n-1}(t \mid \boldsymbol{t}^{(0)}, \boldsymbol{t}^{(n)})$ as presented in Figure 1. Thereby, we achieve a natural way to generate sequences with varying numbers of events and expressively model arbitrary TPPs. In short, our contributions are as follows:

- We connect diffusion models with TPPs by introducing a novel model that naturally handles the discrete and continuous nature of point processes.

- We propose a model that is flexible and permits parallel and closed-form sampling of entire event sequences, overcoming common limitations of autoregressive models.

- We show that our model matches the performance of state-of-the-art TPP models in density estimation and outperforms them in forecasting.

## 2 Background

### 2.1 Temporal point processes (TPPs)

Temporal point processes (TPPs) [8, 9] are stochastic processes that define a probability distribution over event sequences whose number of points (events) $K$ and their locations (arrival times) $t_i$ are random. A realization of a TPP can be represented as a sequence of strictly increasing arrival times: $\boldsymbol{t} = (t_1, \ldots, t_K)$, $0 < t_1 < \cdots < t_K \leq T$. Viewing a TPP as a counting process, we can equivalently represent a TPP realization by a counting measure $N(t) = \sum_i^K \mathbb{I}(t_i \leq t)$, for $t \in [0, T]$. The intensity characterizing a TPP can be interpreted as the expected number of events per unit of time and is defined as:

$$\lambda(t \mid \mathcal{H}_t) = \lim_{\Delta t \downarrow 0} \frac{\mathbb{E}[N(t + \Delta t) - N(t) \mid \mathcal{H}_t]}{\Delta t}, \tag{1}$$

where $\mathcal{H}_t = \{t_i : t_i < t\}$ is the event history until time $t$, which acts as a filtration to the process.

TPPs have a number of convenient theoretical properties, two of which will be central to our derivation of a noising process for TPPs later in the paper. The first property is *superposition*: If we combine events generated by TPPs with intensities $\lambda_1(t)$ and $\lambda_2(t)$, the resulting event sequence will again follow a TPP, but now with intensity $\lambda_1(t) + \lambda_2(t)$. Conversely, randomly removing each event generated by a TPP process with intensity $\lambda(t)$ with probability $p$ is known as independent *thinning*. This is equivalent to sampling from a TPP with intensity $(1 - p)\lambda(t)$ [9].

**Poisson process.** A *(in)homogeneous* Poisson process is the simplest class of TPPs, where the rate of event occurrence is independent of the history. Then the number of points on $[0, T]$ follows a Poisson distribution with rate $\Lambda(T) = \int_0^T \lambda(t) \, dt$. In the context of our model, the Poisson process with a constant intensity on $[0, T]$, called *homogeneous* Poisson Process (HPP), represents the noise distribution. Even though Poisson processes can model seasonality, i.e., time-varying rates of event occurrence, they assume the events to be independent and do not capture the exciting or inhibiting behavior present in the real world, e.g., a large earthquake increasing the likelihood of observing other earthquakes soon after.

**Conditional intensity.** Most TPP models leverage the *conditional intensity* function $\lambda(t \mid \mathcal{H}_t)$ or equivalently the conditional density $p(t \mid \mathcal{H}_t)$ to overcome the independence of points limitation of an *inhomogeneous* Poisson process. Historically, these intensity models were parameterized using hand-crafted functions [17, 20], whereas now, it is more common to use neural networks for learning intensities from data [12, 33, 41]. While the conditional intensity provides a general framework to model TPPs, sampling from these models is inherently autoregressive.

## 2.2 Denoising diffusion probabilistic models

Diffusion models [18, 45] are a class of latent variable models that learn a generative model to reverse a fixed probabilistic noising process $\boldsymbol{x}_0 \to \boldsymbol{x}_1 \to \cdots \to \boldsymbol{x}_N$, which gradually adds noise to clean data $\boldsymbol{x}_0$ until no information remains, i.e., $\boldsymbol{x}_N \sim p(\boldsymbol{x}_N)$. For continuous data, the forward (noising) process is usually defined as a fixed Markov chain $q(\boldsymbol{x}_n \mid \boldsymbol{x}_{n-1})$ with Gaussian transitions. Then the Markov chain of the reverse process is captured by approximating the true posterior $q(\boldsymbol{x}_{n-1} \mid \boldsymbol{x}_0, \boldsymbol{x}_n)$ with a model $p_\theta(\boldsymbol{x}_{n-1} \mid \boldsymbol{x}_n)$. Ultimately, sampling new realizations $\boldsymbol{x}_0$ from the modeled data distribution $p_\theta(\boldsymbol{x}_0) = \int p(\boldsymbol{x}_N) \prod_{n=1}^N p_\theta(\boldsymbol{x}_{n-1} \mid \boldsymbol{x}_n) \, \mathrm{d}\boldsymbol{x}_1 \ldots \boldsymbol{x}_N$ is performed by starting with a sample from pure noise $\boldsymbol{x}_N \sim p(\boldsymbol{x}_N)$ and gradually denoising it with the learned model over $N$ steps $\boldsymbol{x}_N \to \boldsymbol{x}_{N-1} \to \cdots \to \boldsymbol{x}_0$.

## 3  ADD-THIN

In the following, we derive a diffusion-like model for TPPs—ADD-THIN. The two main components of this model are the forward process, which converts data to noise (noising), and the reverse process, which converts noise to data (denoising). We want to emphasize again that existing Gaussian diffusion models [18, 45] are not suited to model entire event sequences, given that the number of events is random and the arrival times are strictly positive. For this reason, we will derive a new noising and denoising process (Sections 3.1 & 3.2), present a learnable parametrization and appropriate loss to approximate the posterior (Section 3.3) and introduce a sampling procedure (Sections 3.4 & 3.5).

## 3.1  Forward process – Noising

Let $\boldsymbol{t}^{(0)} = (t_1, \ldots, t_K)$ denote an i.i.d. sample from a TPP (data process) with $T = 1$ specified by the unknown (conditional) intensity $\lambda_0$. We define a *forward* noising process as a sequence of TPPs that start with the true intensity $\lambda_0$ and converge to a standard HPP, i.e., $\lambda_0 \to \lambda_1 \to \cdots \to \lambda_N$:

$$\lambda_n(t) = \underbrace{\alpha_n \lambda_{n-1}(t)}_{\text{(i) Thin}} + \underbrace{(1 - \alpha_n)\lambda_{\text{HPP}}}_{\text{(ii) Add}}, \tag{2}$$

where $1 > \alpha_1 > \alpha_2 > \cdots > \alpha_N > 0$ and $\lambda_{\text{HPP}}$ denotes the constant intensity of an HPP. Equation 2 corresponds to a superposition of (i) a process $\lambda_{n-1}$ thinned with probability $1 - \alpha_n$ (removing old points), and (ii) an HPP with intensity $(1 - \alpha_n)\lambda_{\text{HPP}}$ (adding new points).

**Property** (Stationary intensity). *For any starting intensity $\lambda_0$, the intensity function $\lambda_N$ given by Equation 2 converges towards $\lambda_{\text{HPP}}$. That is, the noised TPP will be an HPP with $\lambda_{\text{HPP}}$.*

*Proof.* In Appendix B.1 we show that, given $\lambda_0$ and Equation 2, $\lambda_n$ is given by:

$$\lambda_n(t) = \bar{\alpha}_n \lambda_0(t) + (1 - \bar{\alpha}_n)\lambda_{\text{HPP}}, \tag{3}$$

where $\bar{\alpha}_n = \prod_j^n \alpha_j$. Since $\prod_j^N \alpha_j \to 0$ as $N \to \infty$, thus $\lambda_N(t) \to \lambda_{\text{HPP}}$. $\qquad\square$

If all $\alpha_n$ are close to 1, each consecutive realization will be close to the one before because we do not remove a lot of original points, nor do we add many new points. And if we have enough steps, we will almost surely converge to the HPP. Both of these properties will be very useful in training a generative model that iteratively reverses the forward noising process.

But how can we sample points from $\lambda_n$ if we do not have access to $\lambda_0$? Since we know the event sequence $\boldsymbol{t}^{(0)}$ comes from a true process which is specified with $\lambda_0$, we can sample from a thinned process $\bar{\alpha}_n \lambda_0(t)$, by thinning the points $\boldsymbol{t}^{(0)}$ independently with probability $1 - \bar{\alpha}_n$. This shows that even though we cannot access $\lambda_0$, we can sample from $\lambda_n$ by simply thinning $\boldsymbol{t}^{(0)}$ and adding new points from an HPP.

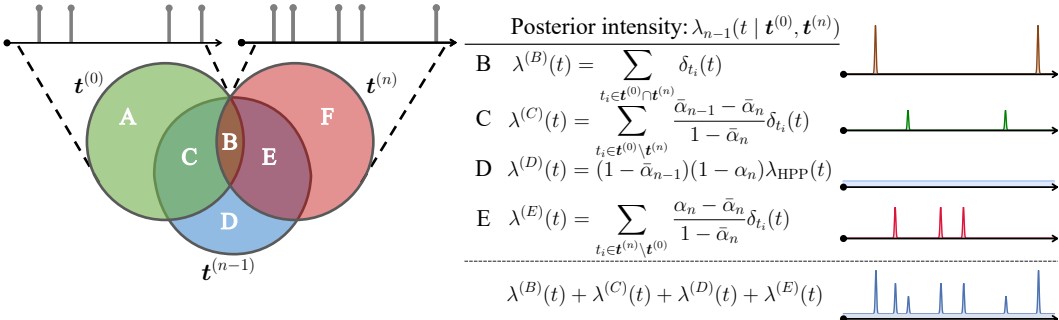

Figure 2: *(Left)* Illustration of all possible disjoint sets that we can reach in our forward process going from $t^{(0)}$ to $t^{(n)}$ through $t^{(n-1)}$. *(Right)* Posterior intensity describing the distribution of $t^{(n-1)} \mid t^{(0)}, t^{(n)}$, where each subset **B**-**E** can be generated by sampling from the intensity functions.

To recap, given a clean sequence $t^{(0)}$, we obtain progressively noisier samples $t^{(n)}$ by both removing original points from $t^{(0)}$ and adding new points at random locations. After $N$ steps, we reach a sample corresponding to an HPP—containing no information about the original data.

## 3.2   Reverse process – Denoising

To sample realizations $t \sim \lambda_0$ starting from $t^{(N)} \sim \lambda_{HPP}$, we need to learn to reverse the Markov chain of the forward process, i.e., $\lambda_N \to \cdots \to \lambda_0$, or equivalently $t^{(N)} \to \cdots \to t^{(0)}$. Conditioned on $t^{(0)}$, the reverse process at step $n$ is given by the posterior $q(t^{(n-1)} \mid t^{(0)}, t^{(n)})$, which is an inhomogeneous Poisson process for the chosen forward process (Section 3.1). Therefore, the posterior can be represented by a history-independent intensity function $\lambda_{n-1}(t \mid t^{(0)}, t^{(n)})$.

As the forward process is defined by adding and thinning event sequences, the points in the random sequence $t^{(n-1)}$ can be decomposed into disjoint sets of points based on whether they are also in $t^{(0)}$ or $t^{(n)}$. We distinguish the following cases: points in $t^{(n-1)}$ that were kept from 0 to $n$ (**B**), points in $t^{(n-1)}$, that were kept from 0 to $n-1$ but thinned at the $n$-th step (**C**), added points in $t^{(n-1)}$ that are thinned in the $n$-th step (**D**) and added points in $t^{(n-1)}$ that are kept in the $n$-th step (**E**). Since the sets **B**-**E** are disjoint, the posterior intensity is a superposition of the intensities of each subsets of $t^{(n-1)}$: $\lambda_{n-1}(t \mid t^{(0)}, t^{(n)}) = \lambda^{(B)}(t) + \lambda^{(C)}(t) + \lambda^{(D)}(t) + \lambda^{(E)}(t)$.

To derive the intensity functions for cases **B**-**E**, we additionally define the following helper sets: **A** the points $t^{(0)} \setminus t^{(n-1)}$ that were thinned until $n-1$ and **F** the points $t^{(n)} \setminus t^{(n-1)}$ that have been added at step $n$. The full case distinction and derived intensities are further illustrated in Figure 2. In the following paragraphs, we derive the intensity functions for cases **B**-**E**:

**Case B:** The set $t^{(B)}$ can be formally defined as $t^{(0)} \cap t^{(n)}$ since $(t^{(0)} \cap t^{(n)}) \setminus t^{(n-1)} = \emptyset$ almost surely. This is because adding points at any of the locations $t \in t^{(0)} \cap t^{(n)}$ carries zero measure at every noising step. Hence, given $t^{(0)} \cap t^{(n)}$ the intensity can be written as a sum of Dirac measures: $\lambda^{(B)}(t) = \sum_{t_i \in (t^{(0)} \cap t^{(n)})} \delta_{t_i}(t)$. Similar to how the forward process generated $t^{(B)}$ by preserving some points from $t^{(0)}$, sampling from the reverse process preserves points from $t^{(n)}$.

**Case C:** Given $t^{(A \cup C)} = t^{(0)} \setminus t^{(n)}$, $t^{(C)}$ can be found by thinning and consists of points that were kept by step $n-1$ and removed at step $n$. Using the thinning of Equations 2 and 3, we know the probability of a point from $t^{(0)}$ being in $t^{(C)}$ and $t^{(A \cup C)}$ is $\bar{\alpha}_{n-1}(1 - \alpha_n)$ and $1 - \bar{\alpha}_n$, respectively. Since we already know $t^{(B)}$ we can consider the probability of finding a point in $t^{(C)}$, given $t^{(A \cup C)}$, which is equal to $\frac{\bar{\alpha}_{n-1} - \bar{\alpha}_n}{1 - \bar{\alpha}_n}$. Consequently, $\lambda^{(C)}(t)$ is given as a thinned sum of Dirac measures over $t^{(A \cup C)}$ (cf., Figure 2).

**Case D:** The set $t^{(D)}$ contains all points $t \notin (t^{(0)} \cup t^{(n)})$ that were added until step $n-1$ and thinned at step $n$. Again using Equations 2 and 3, we can see that these points were added with intensity $(1 - \bar{\alpha}_{n-1})\lambda_{HPP}$ and then removed with probability $\alpha_n$ at the next step. Equivalently, we can write down the intensity that governs this process as $\lambda^{(D)}(t) = (1 - \bar{\alpha}_{n-1})(1 - \alpha_n)\lambda_{HPP}$, i.e., sample points from an HPP and thin them to generate a sample $t^{(D)}$.

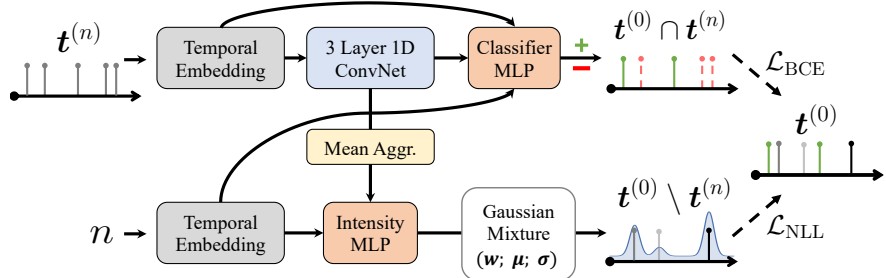

Figure 3: Architecture of our model predicting $t_0$ from $t_n$.

**Case E:** The set $t^{(E)}$ can be found by thinning $t^{(E \cup F)} = t^{(n)} \setminus t^{(0)}$ and contains the points that were added by step $n-1$ and then kept at step $n$. The processes that generated $t^{(E)}$ and $t^{(F)}$ are two independent HPPs with intensities $\lambda^{(E)} = (1 - \bar{\alpha}_{n-1}) \alpha_n \lambda_{HPP}$ and $\lambda^{(F)} = (1 - \alpha_n) \lambda_{HPP}$, respectively, where $\lambda^{(E)}(t)$ is derived in a similar way to $\lambda^{(D)}(t)$. Since $t^{(E)}$ and $t^{(F)}$ are independent HPPs and we know $t^{(E \cup F)}$, the number of points in $t^{(E)}$ follows a Binomial distribution with probability $p = \frac{\lambda^{(E)}}{\lambda^{(E)} + \lambda^{(F)}}$ (see Appendix B.2 for details). That means we can sample $t^{(E)}$ given $t^{(n)}$ and $t^{(0)}$ by simply thinning $t^{(E \cup F)}$ with probability $1-p$ and express the intensity as a thinned sum of Dirac functions (cf., Figure 2).

For sequences of the training set, where $t^{(0)}$ is known, we can compute these intensities for all samples $t^{(n)}$ and reverse the forward process. However, $t^{(0)}$ is unknown when sampling new sequences. Therefore, similarly to the denoising diffusion approaches [18], in the next section, we show how to approximate the posterior intensity, given only $t^{(n)}$. Further, in Section 3.4, we demonstrate how the trained neural network can be leveraged to sample new sequences $t \sim \lambda_0$.

### 3.3 Parametrization and training

In the previous section we have derived the intensity $\lambda_{n-1}(t \mid t^{(0)}, t^{(n)})$ of the posterior $q(t^{(n-1)} \mid t^{(0)}, t^{(n)})$ for the reverse process, i.e., the intensity of points at step $n-1$ given $t^{(n)}$ and $t^{(0)}$. Now we want to approximate this posterior using a model $p_\theta(t^{(n-1)} \mid t^{(n)}, n)$ to learn to sample points $t^{(n-1)}$ given only $t^{(n)}$. As we are only missing information about $t^{(0)}$ we will learn to model $\lambda^{(B)}(t)$ and $\lambda^{(A \cup C)}(t)$ to approximate $\hat{t}^{(0)} \approx t^{(0)}$ (cf., Figure 3) for each $n$ and then have access to the full posterior intensity from Section 3.2 to reverse the noising process.

**Sequence embedding.** To condition our model on $t^{(n)}$ and $n$ we propose the following embeddings. We use a sinusoidal embedding [47] to embed the diffusion time $n$. Further, we encode each arrival time $t_i \in t^{(n)}$ and inter-event time $\tau_i = t_i - t_{i-1}$, with $t_0 = 0$, to produce a temporal event embedding $e_i \in \mathbb{R}^d$ by applying a sinusoidal embedding [47]. Then we leverage a three-layered 1D-Convolutional Neural Network (CNN) with circular padding, dilation, and residual connections to compute a context embedding $c_i \in \mathbb{R}^d$. Compared to attention and RNN-based encoders, the CNN is computationally more efficient and scales better with the length of event sequences while allowing us to capture long-range dependencies between the events. Finally, a global sequence embedding $\bar{c}$ is generated by a mean aggregation of the context embeddings.

**Posterior approximation.** The posterior defining $t^{(D)}$ is independent of $t^{(0)}$ and $t^{(n)}$ and can be sampled from directly. The set $t^{(B)}$ corresponds to those points that were kept from $t^{(0)}$ until the $n$-th step. Since all of these points are also included in $t^{(n)}$ we specify a classifier $g_\theta(e_i, c_i, n)$ with an MLP that predicts which points from $t^{(n)}$ belong to $t^{(0)}$. As $t^{(0)}$ is known during training, this is a standard classification setting. We use the binary cross entropy (BCE) loss $\mathcal{L}_{\text{BCE}}$ to train $g_\theta$. Note that classifying $t^{(B)}$ from $t^{(n)}$ simultaneously predicts $t^{(n)} \setminus t^{(0)} = t^{(E \cup F)}$. Therefore we can subsequently attain $t^{(E)}$ by thinning $t^{(E \cup F)}$ as explained in Section 3.2.

To sample points $t^{(C)}$ we have to specify the intensity function $\lambda^{(A \cup C)}(t)$ that will be thinned to attain $t^{(C)}$ (cf., Section 3.2). As $\lambda^{(A \cup C)}(t)$ is a mixture of Dirac functions we use an *unnormalized* mixture of $H$ weighted and truncated Gaussian density functions $f$ on $[0, T]$ to parameterize the

inhomogeneous intensity:

$$\lambda_\theta^{(A\cup C)}(t) = K \sum_{j=1}^{H} w_j f\left(t; \mu_j, \sigma_j\right),\tag{4}$$

where $w_j = \mathrm{Softplus}(\mathrm{MLP}_w([n,\overline{c}]))$, $\mu_j = \mathrm{Sigmoid}(\mathrm{MLP}_\mu([n,\overline{c}]))$ and $\sigma_j = \exp(-|\mathrm{MLP}_\sigma([n,\overline{c}])|)$ are parameterized by MLPs with two layers, a hidden dimension of $d$ and a ReLU activation. Note that the Gaussian function is the standard approximation of the Dirac delta function and can, in the limit $\sigma \to 0$, perfectly approximate it. We include $K$, the number of points in $t^{(n)}$, in the intensity to more directly model the number of events. Then $\lambda_\theta^{(A\cup C)}(t)$ is trained to match samples $t^{(A\cup C)} \sim \lambda^{(A\cup C)}$ by minimizing the negative log-likelihood (NLL):

$$\mathcal{L}_{\mathrm{NLL}} = -\log p(t^{(A\cup C)}) = -\sum_{t_i \in t^{(A\cup C)}} \log \lambda_\theta^{(A\cup C)}(t_i) + \int_0^T \lambda_\theta^{(A\cup C)}(t)\,\mathrm{d}t.\tag{5}$$

Thanks to the chosen parametrization, the integral term in $\mathcal{L}_{\mathrm{NLL}}$ can be efficiently computed in any deep-learning framework using the 'erf' function, without relying on Monte Carlo approximation. We present an overview of our model architecture to predict $t^{(0)}$ from $t^{(n)}$ in Figure 3.

**Training objective.** The full model is trained to minimize $\mathcal{L} = \mathcal{L}_{\mathrm{NLL}} + \mathcal{L}_{\mathrm{BCE}}$. During training, we do not have to evaluate the true posterior or sample events from any of the posterior distributions. Instead, we can simply sample $n$ and subsequently $t^{(n)}$ and minimize $\mathcal{L}$ for $t^{(n)}$. Interestingly, in Appendix A, we show that $\mathcal{L}$ is equivalent to the Kullback-Leibler (KL) divergence between the approximate posterior $p_\theta(t^{(n-1)} \mid t^{(n)}, n)$ and the true posterior $q(t^{(n-1)} \mid t^{(0)}, t^{(n)})$. Ultimately, this shows that optimizing the evidence lower bound (ELBO) of the proposed model boils down to simply learning a binary classification and fitting an inhomogeneous intensity.

## 3.4 Sampling

To sample an event sequence from our model, we start by sampling $t^{(N)}$ from an HPP with $\lambda_{\mathrm{HPP}}$. Subsequently, for each $n \in [N, \cdots, 1]$, $\hat{t}^{(0)}$ is predicted by classifying $\hat{t}^{(B)}$ and sampling $\hat{t}^{(A\cup C)}$ from $\lambda_\theta^{(A\cup C)}(t)$. Note that the Poisson distribution with intensity $\Lambda^{(A\cup C)}(T) = \int_0^T \lambda_\theta^{(A\cup C)}(t)\,\mathrm{d}t$ parameterizes the number of points in $A \cup C$. Therefore, $\hat{t}^{(A\cup C)}$ can be sampled by first sampling the number of events and then sampling the event times from the normalized intensity $\lambda_\theta^{(A\cup C)}(t)/\Lambda^{(A\cup C)}(T)$. Given our predicted $\hat{t}^{(0)}$ we can sample $\hat{t}^{(n-1)}$ from the posterior intensity defined in Section 3.2. By repeating this process, we produce a sequence of $t^{(n-1)}$s. Finally, we obtain a denoised sample $t^{(0)}$ by predicting it from $\hat{t}^{(1)}$. We provide an overview of the sampling procedure as pseudo-code in Algorithm 1.

---
**Algorithm 1:** Sampling

---
$t^{(n=N)} \sim \lambda_{\mathrm{HPP}}$;
**for** $n \in \{N, \dots, 1\}$ **do**
    sample $\hat{t}^{(B)} \sim \sum_{t_i \in t^{(n)}} g_\theta(t_i \mid e_i, c_i, n)\delta_{t_i}(t)$;
    sample $\hat{t}^{(C)} \sim \frac{\bar\alpha_{n-1}-\bar\alpha_n}{1-\bar\alpha_n}\lambda_\theta^{(A\cup C)}(t \mid t^{(n)}, n)$;
    sample $\hat{t}^{(D)} \sim (1-\bar\alpha_{n-1})(1-\alpha_n)\lambda_{\mathrm{HPP}}$;
    sample $\hat{t}^{(E)} \sim \sum_{t_i \in t^{(n)}\setminus \hat{t}^{(B)}} \frac{\alpha_n-\bar\alpha_n}{1-\bar\alpha_n}\delta_{t_i}(t)$;
    $t^{(n-1)} \leftarrow \hat{t}^{(B)} \cup \hat{t}^{(C)} \cup \hat{t}^{(D)} \cup \hat{t}^{(E)}$;
**end**
sample $\hat{t}^{(A\cup C)} \sim \lambda_\theta^{(A\cup C)}(t \mid t^{(1)}, 1)$;
sample $\hat{t}^{(B)} \sim \sum_{t_i \in t^{(1)}} g_\theta(t_i \mid e_i, c_i, 1)\delta_{t_i}(t)$;
$t \leftarrow \hat{t}^{(B)} \cup \hat{t}^{(A\cup C)}$;
**return** $t$

---

## 3.5 Conditional sampling

The above-described process defines an *unconditional* generative model for event sequences on an interval $[0, T]$. For many (multi-step) forecasting applications, such as earthquake forecasting [10], we need to condition our samples on previous event sequences and turn our model into a conditional one that can generate future event sequences in $[H, H + T]$ given the past observed events in $[0, H]$. To condition our generative model on a history, we apply a simple GRU encoder to encode the history into a $d$-dimensional history embedding $h$, which subsequently conditions the classifier and intensity model by being added to the diffusion time embedding.

# 4 Related work

**Autoregressive neural TPPs.**  Most neural TPPs model the intensity or density of each event conditional on a history and consequently consist of two parts: a history encoder and an intensity/density decoder. As history encoders, RNNs [12, 41] and attention-based set encoders [50, 52] have been proposed. Attention-based encoders are postulated to better model long-range dependencies in the event sequences, but at the cost of a more complex encoder structure [43]. To decode the intensity $\lambda(t|\mathcal{H})$, density $p(t|\mathcal{H})$ or the cumulative hazard function from the history, uni-modal distributions [12], mixture-distributions [41], a mixture of kernels [36, 44, 51], neural networks [37] and Gaussian diffusion [29] have been proposed. Another branch of neural TPPs models the event times conditional on a latent variable that follows a continuous-time evolution [5, 13, 16, 21], where, e.g., Hasan et al. [16] relate inter-event times of a TPP to the excursion of a stochastic process. In general, most neural TPPs are trained by maximizing the log-likelihood, but other training approaches have been proposed [26, 29, 49]. We want to highlight the difference of our model to two related works. TriTPP [42] learns a deterministic mapping between a latent HPP and a TPP using normalizing flows, which allows for parallel sampling. However, it models the conditional hazard function, which forces a conditional dependency of later arrival times and can still produce error accumulation. Lin et al. [29] proposed an autoregressive TPP model leveraging Gaussian diffusion to approximate the conditional density. Besides being autoregressive, the model does not directly model the number of points in the TPP but instead is trained to maximize the ELBO of the next inter-event time.

**Non-autoregressive neural TPPs.**  An alternative to the conditional (autoregressive) modeling of TPPs is to apply a latent variable model that learns to relate entire point processes to latent variables. The class of Cox processes [7, 11, 19] models point processes through a hierarchy of latent processes under the assumption that higher-level latent variables trigger lower-level realizations. ADD-THIN can be considered to be a non-autoregressive latent variable model.

**Denoising diffusion models.**  Recently, denoising diffusion models on continuous state spaces [18, 45] established the new state-of-the-art for many image applications [15, 18, 25]. Subsequently, diffusion models for other application domains such as point clouds [31, 32], physical simulations [23, 28, 30] and time-series [1, 3, 24, 46] emerged. While the majority of denoising diffusion models are based on Gaussian transition kernels in continuous state spaces proposed in [18, 45], a variety of diffusion models for discrete state spaces such as graphs [48], text [2, 27] and images [2, 6] have been presented. Here, we highlight the similarity of our work to the concurrent work of Chen and Zhou [6], who derived a diffusion process that models pixel values as a count distribution and thins them to noise images. In contrast to the related work on continuous and discrete state space diffusion models, ADD-THIN constitutes a novel diffusion model defined on a state space that captures both the discrete and continuous components of point processes.

# 5 Experiments

We evaluate the proposed model in two settings: density estimation and forecasting. In density estimation, the goal is to learn an unconditional model for event sequences. As for forecasting, the objective is to accurately predict the entire future event sequence given the observed past events.

**Data.**  ADD-THIN is evaluated on 7 real-world datasets proposed by Shchur et al. [42] and 6 synthethic datasets from Omi et al. [37]. The synthetic datasets consist of Hawkes1 and Hawkes2 [17], a self-correcting (SC) [20], inhomogeneous Poisson process (IPP) and a stationary and a non-stationary renewal process (MRP, RP) [39, 11]. For the real-world datasets, we consider PUBG, Reddit-Comments, Reddit-Submissions, Taxi, Twitter, and restaurant check-ins in Yelp1 and Yelp2. We split each dataset into train, validation, and test set containing 60%, 20%, and 20% of the event sequences, respectively. Further dataset details and statistics are reported in Appendix C.

**Baselines.**  We apply the *RNN*-based intensity-free TPP model from Shchur et al. [41]. Similar to Sharma et al. [40], we further combine the intensity-free model with an attention-based encoder from Zuo et al. [52] as a *Transformer* baseline. Additionally, we compare our model to an autoregressive TPP model with a continuous state Gaussian-diffusion [18] from Lin et al. [29], which we abbreviate as *GD*. Lastly, *TriTPP* [42] is used as a model that provides parallel but autoregressive sampling.

Table 1: MMD ($\downarrow$) between the TPP distribution of sampled sequences and hold-out test set (**bold** best, underline second best). The results with standard deviation are reported in Appendix E.

| | Hawkes1 | Hawkes2 | SC | IPP | RP | MRP | PUBG | Reddit-C | Reddit-S | Taxi | Twitter | Yelp1 | Yelp2 |
|---|---|---|---|---|---|---|---|---|---|---|---|---|---|
| RNN | **0.02** | **0.01** | **0.08** | 0.05 | **0.01** | **0.03** | 0.04 | **0.01** | **0.02** | **0.04** | **0.03** | 0.07 | **0.03** |
| Transformer | 0.03 | 0.04 | 0.19 | 0.10 | 0.02 | 0.19 | 0.06 | 0.05 | 0.09 | 0.09 | 0.08 | 0.12 | 0.14 |
| GD | 0.06 | 0.06 | 0.13 | 0.08 | 0.05 | 0.14 | 0.11 | 0.03 | 0.03 | 0.10 | 0.15 | 0.12 | 0.10 |
| TriTPP | 0.03 | 0.04 | 0.23 | 0.04 | 0.02 | 0.05 | 0.06 | 0.09 | 0.12 | 0.07 | 0.04 | **0.06** | 0.06 |
| ADD-THIN | **0.02** | 0.02 | 0.19 | **0.03** | 0.02 | 0.10 | **0.03** | **0.01** | **0.02** | **0.04** | 0.04 | 0.08 | 0.04 |

Table 2: Wasserstein distance ($\downarrow$) between the distribution of the number of events of sampled sequences and hold-out test set (**bold** best, underline second best). The results with standard deviation are reported in Appendix E.

| | Hawkes1 | Hawkes2 | SC | IPP | RP | MRP | PUBG | Reddit-C | Reddit-S | Taxi | Twitter | Yelp1 | Yelp2 |
|---|---|---|---|---|---|---|---|---|---|---|---|---|---|
| RNN | **0.03** | **0.01** | **0.00** | 0.02 | 0.02 | **0.01** | **0.02** | **0.01** | 0.05 | **0.02** | **0.01** | 0.04 | **0.02** |
| Transformer | 0.06 | 0.04 | 0.06 | 0.07 | 0.04 | 0.11 | 0.04 | 0.08 | 0.11 | 0.13 | 0.05 | 0.11 | 0.21 |
| GD | 0.16 | 0.13 | 0.50 | 0.42 | 0.28 | 0.50 | 0.54 | 0.02 | 0.16 | 0.33 | 0.07 | 0.26 | 0.25 |
| TriTPP | **0.03** | 0.03 | 0.01 | **0.01** | 0.02 | 0.03 | 0.03 | 0.09 | 0.09 | 0.04 | **0.01** | **0.03** | 0.04 |
| ADD-THIN | 0.04 | 0.02 | 0.08 | **0.01** | 0.02 | 0.04 | **0.02** | 0.03 | **0.04** | 0.03 | **0.01** | 0.04 | **0.02** |

**Training and model selection.** We train each model by its proposed training loss using Adam [22]. For our model, we set the number of diffusion steps to $N = 100$, apply the cosine beta-schedule proposed in Glide [34], and set $\lambda_{\text{HPP}} = 1$ for the noising process. We apply early stopping, hyperparameter tuning, and model selection on the validation set for each model. Further hyperparameter and training details are reported in Appendix D.

## 5.1 Sampling – Density estimation

A good TPP model should be flexible enough to fit event sequences from various processes. We evaluate the generative quality of the TPP models on 13 synthetic and real-world datasets by drawing 4000 TPP sequences from each model and computing distance metrics between the samples and event sequences from a hold-out test set. In short, the goal of this experiment is to show that our proposed model is a flexible TPP model that can generate event sequences that are 'close' to the samples from the data-generating distribution.

**Metrics.** In general, diffusion models cannot evaluate the exact likelihood. Instead, we evaluate the quality of samples by comparing the distributions of samples from each model and the test set with the maximum mean discrepancy measure (MMD) [14] as proposed by Shchur et al. [42]. Furthermore, we compute the Wasserstein distance [38] between the distribution of sequence lengths of the sampled sequences and the test set. We report the results on the test set averaged over five runs with different seeds.

**Results.** Table 1 presents the MMD results for all models and datasets. Among them, the RNN baseline demonstrates a strong performance across all datasets and outperforms both *Transformer* and *GD*. Notably, ADD-THIN exhibits competitive results with the autoregressive baseline, surpassing or matching ($\pm 0.01$) it on 11/13 datasets. Additionally, ADD-THIN consistently outperforms the *Transformer* and *GD* model on all datasets except SC and RP. Lastly, *TriTPP* performs well on most datasets but is outperformed or matched by our model on all but two datasets.

Table 2 shows the result for comparing the count distributions. Overall, the Wasserstein distance results align closely with the MMD results. However, the *GD* model is an exception, displaying considerably worse performance when focusing on the count distribution. This outcome is expected since the training of the *GD* model only indirectly models the number of events by maximizing the ELBO of each diffusion step to approximate the conditional density of the next event and not the likelihood of whole event sequences. Again, ADD-THIN shows a very strong performance, further emphasizing its expressiveness.

In summary, these results demonstrate the flexibility of our model, which can effectively capture various complex TPP distributions and matches the state-of-the-art performance in density estimation.

Table 3: Wasserstein distance between forecasted event sequence and ground truth reported for 50 random forecast windows on the test set (lower is better). The results with standard deviation are reported in Appendix E.

| | PUBG | Reddit-C | Reddit-S | Taxi | Twitter | Yelp1 | Yelp2 |
|---|---|---|---|---|---|---|---|
| Average Seq. Length | 76.5 | 295.7 | 1129.0 | 98.4 | 14.9 | 30.5 | 55.2 |
| RNN | 6.15 | 35.22 | 39.23 | 4.14 | 2.04 | 1.28 | 2.21 |
| Transformer | 2.45 | 38.77 | 27.52 | 3.12 | 2.09 | 1.29 | 2.64 |
| GD | 5.44 | 44.72 | 64.25 | 4.32 | 2.16 | 1.52 | 4.25 |
| ADD-THIN (Ours) | **2.03** | **17.18** | **21.32** | **2.42** | **1.48** | **1.00** | **1.54** |

Table 4: Count MAPE $\times 100\%$ between forecasted event sequences and ground truth reported for 50 random forecast windows on the test set (lower is better). The results with standard deviation are reported in Appendix E.

| | PUBG | Reddit-C | Reddit-S | Taxi | Twitter | Yelp1 | Yelp2 |
|---|---|---|---|---|---|---|---|
| Average Seq. Length | 76.5 | 295.7 | 1129.0 | 98.4 | 14.9 | 30.5 | 55.2 |
| RNN | 1.72 | 5.47 | 0.68 | 0.54 | 0.95 | 0.59 | 0.72 |
| Transformer | 0.65 | 7.38 | 0.55 | 0.46 | 1.18 | 0.63 | 0.99 |
| GD | 1.66 | 10.49 | 1.33 | 0.71 | 1.43 | 0.78 | 1.65 |
| ADD-THIN (Ours) | **0.45** | **1.07** | **0.38** | **0.37** | **0.69** | **0.45** | **0.50** |

## 5.2 Conditional sampling – Forecasting

Forecasting event sequences from history is an important real-world application of TPP models. In this experiment, we evaluate the forecasting capability of our model on all real-world datasets. We evaluate each model's performance in forecasting the events in a forecasting window $\Delta T$, by randomly drawing a starting point $T_s \in [\Delta T, T - \Delta T]$. Then, the events in $[0, T_s]$ are considered the history, and $[T_s, T_s + \Delta T]$ is the forecasting time horizon.

In the experiment, we randomly sample 50 forecasting windows for each sequence from the test set, compute history embedding with each model's encoder, and then conditionally sample the forecast from each model. Note that *TriTPP* does not allow for conditional sampling and is therefore not part of the forecasting experiment.

**Metrics.** To evaluate the forecasting experiment, we will not compare distributions of TPPs but rather TPP instances. We measure the distance between two event sequences, i.e., forecast and ground truth data, by computing the distance between the count measures with the Wasserstein distance between two TPPs, as introduced by Xiao et al. [49]. Additionally, we report the mean absolute relative error (MAPE) between the predicted sequence length and ground truth sequence length in the forecasting horizon. We report the results on the test set averaged over five runs with different seeds.

**Results.** Table 3 presents the average Wasserstein distance between the predicted and ground truth forecast sequences. The results unequivocally demonstrate the superior performance of our model by forecasting entire event sequences, surpassing all autoregressive baselines on all datasets. Notably, the disparity between ADD-THIN and the baselines is more pronounced for datasets with a higher number of events per sequence, indicating the accumulation of prediction errors in the autoregressive models. Further, the transformer baseline achieves better forecasting results than the RNN baseline for some datasets with more events. This suggests that long-range attention can improve autoregressive forecasting and mitigate some error accumulation.

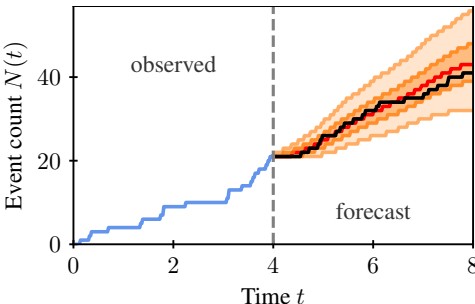

Figure 4: 5%, 25%, 50%, 75%, and 95% quantile of forecasts generated by ADD-THIN for a Taxi event sequence (*blue*: history, *black* ground truth future).

Table 4 reports the MAPE between the forecasted and ground truth sequence length. The MAPE results align consistently with the Wasserstein distance across all models and datasets. Figure 4

depicts the quantiles for 1000 forecasts generated by ADD-THIN for one Taxi event sequence and highlights the predictive capacity of our model. Overall, our model outperforms state-of-the-art TPP models in forecasting real-world event sequences.

## 6   Discussion

**ADD-THIN vs. autoregressive TPP models.**    On a conceptual level, ADD-THIN presents a different trade-off compared to other TPP models: Instead of being autoregressive in event time, our model gradually refines the entire event sequence in parallel at every diffusion step to produce a sample from the learned data distribution. Thereby, we have found that our model is better suited for forecasting and modeling very long event sequences than autoregressive TPP models. Furthermore, the iterative refinement of the entire sequence allows us to leverage simple and shared layers to accurately model the long-range interaction between events and results in nearly constant sampling times across different sequence lengths (cf., Appendix E.3).

**Limitations and future work.**    With ADD-THIN, we have derived a novel diffusion-inspired model for TPPs. Thereby, we focused on modeling the arrival times of the events and did not model continuous and discrete marks. However, we see this as an exciting extension to our framework, which might incorporate Gaussian diffusion [18] for continuous marks and discrete diffusion [2] for discrete marks. Further, while generative diffusion is known to produce high-quality samples, it also can be expensive. Besides tuning the number of diffusion steps, future work could focus on alternative and faster sampling routines [35]. Ultimately, we hope that by having connected diffusion models with TPPs, we have opened a new direction to modeling TPPs and broadened the field of diffusion-based models. Here, it would be especially interesting for us to see whether our framework could benefit other application domains in machine learning that involve sets of varying sizes, such as graph generation (molecules), point clouds, and spatial point processes.

## 7   Conclusion

By introducing ADD-THIN, we have connected the fields of diffusion models and TPPs and derived a novel model that naturally handles the discrete and continuous nature of point processes. Our model permits parallel and closed-form sampling of entire event sequences, overcoming common limitations of autoregressive TPP models. In our experimental evaluation, we demonstrated the flexibility of ADD-THIN, which can effectively capture complex TPP distributions and matches the state-of-the-art performance in density estimation. Additionally, in a long-term forecasting task on real-world data, our model distinctly outperforms the state-of-the-art TPP models by predicting entire forecasting windows non-autoregressively.

## Broader impact

We see the proposed model as a general framework to model continuous-time event data. As such, our method can be applied to many fields, where common application domains include traffic, social networks, and electronic health records. We do not find any use cases mentioned above raise ethical concerns; however, it is essential to exercise caution when dealing with sensitive personal data.

## Acknowledgements

This research was supported by the German Research Foundation, grant GU 1409/3-1.

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

# A   On the relationship between the applied loss and the ELBO

In the following section, we investigate the relationship between our applied loss function and the ELBO to the unknown data distribution derived for diffusion models. The ELBO of diffusion models [18] is given as follows:

$$
\mathcal{L}_{ELBO} = \mathbb{E}_q \Big[ \underbrace{D_{KL}\Big(q(\boldsymbol{t}^{(N)} \mid \boldsymbol{t}^{(0)}) \parallel p(\boldsymbol{t}^{(N)})\Big)}_{\mathcal{L}_N} - \underbrace{\log p_\theta(\boldsymbol{t}^{(0)} \mid \boldsymbol{t}^{(1)})}_{\mathcal{L}_0} +
$$
$$
\sum_{n=2}^{N} \underbrace{D_{KL}\Big(q(\boldsymbol{t}^{(n-1)} \mid \boldsymbol{t}^{(0)}, \boldsymbol{t}^{(n)}) \parallel p_\theta(\boldsymbol{t}^{(n-1)} \mid \boldsymbol{t}^{(n)})\Big)}_{\mathcal{L}_n} \Big].
$$

(6)

Additionally, the Janossi density [8] of an event sequence $\boldsymbol{t}$ on $[0, T]$ allows us to represent each element of the ELBO in terms of our derived inhomogeneous (approximate) posterior intensities (Section 3.2 and Figure 2) as follows:

$$
p(\boldsymbol{t}) = \exp\left(-\int_0^T \lambda(t)\,\mathrm{d}t\right) \prod_{t_i \in \boldsymbol{t}} \lambda(t_i).
$$

(7)

$\mathcal{L}_N$: $\mathcal{L}_N$ is constant as the intensity $\lambda_{\mathrm{HPP}}$ defining $q(\boldsymbol{t}^{(N)} \mid \boldsymbol{t}^{(0)})$ and $p(\boldsymbol{t}^{(N)})$ has no learnable parameters.

$\mathcal{L}_0$: We directly train our model to optimize this likelihood term as described in Section 3.3.

$\mathcal{L}_n$: The KL divergence between two densities is defined as:

$$
D_{KL}(q \parallel p_\theta) = \mathbb{E}_q \Big[ \log(q(\boldsymbol{t}^{(n-1)} \mid \boldsymbol{t}^{(0)}, \boldsymbol{t}^{(n)})) - \log(p_\theta(\boldsymbol{t}^{(n-1)} \mid \boldsymbol{t}^{(n)})) \Big],
$$

(8)

where only the right-hand side relies on $\theta$, letting us minimize the KL divergence by maximizing the expectation over the log-likelihood $\log(p_\theta(\boldsymbol{t}^{(n-1)} \mid \boldsymbol{t}^{(n)}))$.

To add some additional context to the KL divergence in $\mathcal{L}_n$ and similar to the derivation of the posterior in Section 3.2, we will further distinguish three cases:

1. **Case B & E:** $q(\boldsymbol{t}^{(n-1)} \mid \boldsymbol{t}^{(0)}, \boldsymbol{t}^{(n)})$ and $p_\theta(\boldsymbol{t}^{(n-1)} \mid \boldsymbol{t}^{(n)})$ are defined by Bernoulli distributions over each element of $\boldsymbol{t}^{(n)}$. By definition the cross-entropy $H(q, p)$ of the distribution $p$ relative to $q$ is given by $H(q, p) = H(q) + D_{KL}(q \parallel p)$, where $H(q)$ is the entropy and $D_{KL}$ the KL divergence. We can see that minimizing the (binary) cross-entropy is equivalent to minimizing the KL divergence, as the entropy $H(q)$ is constant for the data distribution.

2. **Case C:** Minimizing this KL divergence by maximizing the $\mathbb{E}_q \big[ \log(p_\theta(\boldsymbol{t}^{(n-1)} \mid \boldsymbol{t}^{(n)})) \big]$ is proposed in Section 3.2 to learn $\lambda_\theta^{(A \cup C)}(t)$. Note that $q(\boldsymbol{t}^{(n-1)} \mid \boldsymbol{t}^{(0)}, \boldsymbol{t}^{(n)})$ is sampled from by independently thinning $\boldsymbol{t}^{(0)} \setminus \boldsymbol{t}^{(n)}$. Consequently, by minimizing the NLL of our intensity $\lambda_\theta^{(A \cup C)}(t)$ with regards to $\boldsymbol{t}^{(0)} \setminus \boldsymbol{t}^{(n)}$, we optimize the expectation in closed form.

3. **Case D:** Our parametrization uses the same intensity function for $q(\boldsymbol{t}^{(n-1)} \mid \boldsymbol{t}^{(0)}, \boldsymbol{t}^{(n)})$ and $p_\theta(\boldsymbol{t}^{(n-1)} \mid \boldsymbol{t}^{(n)})$, which does not rely on any learned parameters.

# B   Derivations

## B.1   Direct forward sampling

*Proof.* We first repeat Equation 2:

$$
\lambda_n(t) = \alpha_n \lambda_{n-1}(t) + (1 - \alpha_n)\lambda_{\mathrm{HPP}}.
$$

(2)

Then for the first step we can write:

$$
\lambda_1(t) = \alpha_1 \lambda_0(t) + (1 - \alpha_1)\lambda_{\mathrm{HPP}}
$$
$$
= \left(\prod_{j=1}^{n=1} \alpha_j\right) \lambda_0 + \left(1 - \prod_{j=1}^{n=1} \alpha_j\right) \lambda_{\mathrm{HPP}}.
$$

Assuming Equation 3 holds for step $n - 1$:

$$\lambda_{n-1}(t) = \left(\prod_{j=1}^{n-1} \alpha_j\right) \lambda_0(t) + \left(1 - \prod_{j=1}^{n-1} \alpha_j\right) \lambda_{\text{HPP}},$$

we can write for step $n$:

$$\lambda_n(t) = \alpha_n \lambda_{n-1}(t) + (1 - \alpha_n)\lambda_{\text{HPP}}$$

$$= \alpha_n \left(\left(\prod_{j=1}^{n-1} \alpha_j\right) \lambda_0(t) + \left(1 - \prod_{j=1}^{n-1} \alpha_j\right) \lambda_{\text{HPP}}\right) + (1 - \alpha_n)\lambda_{\text{HPP}}$$

$$= \left(\prod_{j=1}^{n} \alpha_j\right) \lambda_0(t) + \left(\alpha_n - \prod_{j=1}^{n} \alpha_j\right) \lambda_{\text{HPP}} + (1 - \alpha_n)\lambda_{\text{HPP}}$$

$$= \left(\prod_{j=1}^{n} \alpha_j\right) \lambda_0(t) + \left(1 - \prod_{j=1}^{n} \alpha_j\right) \lambda_{\text{HPP}}$$

$$= \bar{\alpha}_n \lambda_0(t) + (1 - \bar{\alpha}_n)\lambda_{\text{HPP}},$$

which completes the proof by induction. $\qquad\square$

## B.2 Conditional distribution of Poisson variables

**Proposition.** *Given two independent random variables $X_1 \sim Poisson(\lambda_1)$, $X_2 \sim Poisson(\lambda_2)$, $X_1 \mid X_1 + X_2 = k$ is Binomial distributed, i.e., $X_1 \mid X_1 + X_2 = k \sim Binomial(x_1; k, \frac{\lambda_1}{\lambda_1 + \lambda_1})$.*

*Proof.* The Poisson distributed random variables $X_1$ and $X_2$ have the following joint probability mass function:

$$P(X_1 = x_1, X_2 = x_2) = e^{-(\lambda_1)}\frac{\lambda_1^{x_1}}{x_1!} e^{-\lambda_2}\frac{\lambda_2^{x_2}}{x_2!}, \tag{9}$$

which further defines the joint probability mass function of $P(X_1 = x_1, X_2 = x_2, Y = k)$ if $x_1 + x_2 = k$. Additionally, it is well know that $Y = X_1 + X_2$ is a Poisson random variable with intensity $\lambda_1 + \lambda_2$ and therefore $P(Y = k) = e^{-(\lambda_1 + \lambda_2)}\frac{(\lambda_1 + \lambda_2)^k}{k!}$. Then $P(X_1 = x_1 \mid Y = k)$ is given by the following:

$$P(X_1 = x_1 \mid Y = k) = \frac{P(X_1 = x_1, X_2 = x_2, Y = k)}{P(Y = k)} \text{ if } x_1 + x_2 = k, \tag{10}$$

$$= \frac{e^{-\lambda_1}\frac{\lambda_1^{x_1}}{x_1!} e^{-\lambda_2}\frac{\lambda_2^{x_2}}{x_2!}}{e^{-(\lambda_1 + \lambda_2)}\frac{(\lambda_1 + \lambda_2)^k}{k!}} \text{ if } x_1 + x_2 = k, \tag{11}$$

$$= \frac{k!}{x_1! x_2!} \frac{\lambda_1^{x_1} \lambda_2^{x_2}}{(\lambda_1 + \lambda_2)^k} \text{ if } x_1 + x_2 = k, \tag{12}$$

$$= \frac{k!}{x_1!(k - x_1)!} \frac{\lambda_1^{x_1}}{(\lambda_1 + \lambda_2)^{x_1}} \frac{\lambda_2^{k-x_1}}{(\lambda_1 + \lambda_2)^{k-x_1}} \text{ if } x_1 + x_2 = k, \tag{13}$$

$$= \frac{k!}{x_1!(k - x_1)!} \left(\frac{\lambda_1}{(\lambda_1 + \lambda_2)}\right)^{x_1} \left(1 - \frac{\lambda_1}{(\lambda_1 + \lambda_2)}\right)^{(k-x_1)} \text{ if } x_1 + x_2 = k, \tag{14}$$

where we have leveraged $x_2 = k - x_1$ and $\frac{\lambda_2}{(\lambda_1 + \lambda_2)} = 1 - \frac{\lambda_1}{(\lambda_1 + \lambda_2)}$. As we have shown $P(X_1 = x_1 \mid Y = k)$ follows the Binomial distribution with $p = \frac{\lambda_1}{(\lambda_1 + \lambda_2)}$. $\qquad\square$

Table 5: Statistics for the synthetic datasets.

|  | # Sequences | $T$ | Average sequence length | $\tau$ |
|---|---|---|---|---|
| Hawkes1 | 1000 | 100 | 95.4 | $1.01 \pm 2.38$ |
| Hawkes2 | 1000 | 100 | 97.2 | $0.98 \pm 2.56$ |
| SC | 1000 | 100 | 100.2 | $0.99 \pm 0.71$ |
| IPP | 1000 | 100 | 100.3 | $0.99 \pm 2.22$ |
| RP | 1000 | 100 | 109.2 | $0.83 \pm 2.76$ |
| MRP | 1000 | 100 | 98.0 | $0.98 \pm 1.83$ |

Table 6: Statistics for the real-world datasets.

|  | # Sequences | $T$ | Unit of time | Average sequence length | $\tau$ | $\Delta T$ |
|---|---|---|---|---|---|---|
| PUBG | 3001 | 30 | minutes | 76.5 | $0.41 \pm 0.56$ | 5 |
| Reddit-C | 1356 | 24 | hours | 295.7 | $0.07 \pm 0.28$ | 4 |
| Reddit-S | 1094 | 24 | hours | 1129.0 | $0.02 \pm 0.03$ | 4 |
| Taxi | 182 | 24 | hours | 98.4 | $0.24 \pm 0.40$ | 4 |
| Twitter | 2019 | 24 | hours | 14.9 | $1.26 \pm 2.80$ | 4 |
| Yelp1 | 319 | 24 | hours | 30.5 | $0.77 \pm 1.10$ | 4 |
| Yelp2 | 319 | 24 | hours | 55.2 | $0.43 \pm 0.96$ | 4 |

## C  Datasets

**Synthetic datasets.**  The six synthethic dataset were sampled by Shchur et al. [42] following the procedure in Section 4.1 of Omi et al. [37] and consist of 1000 sequences on the interval $[0, 100]$.

**Real-world datasets.**  The seven real-world datasets were proposed by Shchur et al. [42] and consist of PUBG, Reddit-Comments, Reddit-Submissions, Taxi, Twitter, Yelp1, and Yelp2. The event sequences of **PUBG** represent the death of players in a game of Player Unknown's Battleground (PUBG). The event sequences of **Reddit-Comments** represent the comments on the askscience subreddit within 24 hours after opening the discussion in the period from 01.01.2018 until 31.12.2019. The event sequences of **Reddit-Submissions** represent the discussion submissions on the politics subreddit within a day in the period from 01.01.2017 until 31.12.2019. The event sequences of **Taxi** represent taxi pick-ups in the south of Manhattan, New York. The event sequences of **Twitter** represent tweets by user 25073877. The event sequences of **Yelp1** represent check-ins for the McCarran International Airport recorded for 27 users in 2018. The event sequences of **Yelp2** represent check-ins for all businesses in the city of Mississauga recorded for 27 users in 2018.

We report summary statistics on the datasets in Table 5 and 6. Lately, the validity of some of the widely used real-world benchmark datasets was criticized [4]. In one-step-ahead prediction tasks with teacher forcing, very simple architectures achieved similar results to some of the more advanced ones. However, this seems to be more of a problem of the task than the datasets. In our work, we consider different tasks (density estimation and long-term forecasting) and metrics and have found significant empirical differences between the baselines on these datasets.

## D  Experimental set-up

All models but the transformer baseline were trained on an Intel Xeon E5-2630 v4 @ 2.20 GHz CPU with 256GB RAM and an NVIDIA GeForce GTX 1080 Ti. Given its RAM requirement, the transformer baseline had to be trained with batch size 32 on an NVIDIA A100-PCIE-40GB for the Reddit-C and Reddit-S datasets.

**Hyperparameter tuning.**  has been applied to all models. The hyperparameter tuning was done on the MMD loss between 1000 samples from the model and the validation set. We use a hidden dimension of 32 for all models. Further, we have tuned the learning rate in $\{0.01, 0.001\}$ for all models, the number of mixture components in $\{8, 16\}$ for ADD-THIN, *RNN* and *Transformer*, the number of knots in $\{10, 20\}$ for *TriTPP* and the number of attention layers in $\{2, 3\}$ for the transformer baseline. The values of all other baseline hyperparameters were set to the recommended

values given by the authors. Further, the *GD* baseline has been trained with a batch size of 16, as recommended by the authors. For the forecasting task, we apply the optimal hyperparameters from the density estimation experiment.

**Early-stopping.** Each model has been trained for up to 5000 epochs with early stopping on the MMD metric on the validation set for the density estimation task and on the Wasserstein distance metric on the validation set for the forecasting task.

# E    Additional results

## E.1    Density estimation results with standard deviations

Table 7: **Synthetic data**: MMD ($\downarrow$) between the TPP distribution of sampled sequences and hold-out test set.

|  | Hawkes1 | Hawkes2 | SC | IPP | RP | MRP |
|---|---|---|---|---|---|---|
| RNN | 0.02±0.003 | 0.01±0.002 | 0.08±0.053 | 0.05±0.009 | 0.01±0.001 | 0.03±0.005 |
| Transformer | 0.03±0.011 | 0.04±0.017 | 0.19±0.006 | 0.10±0.034 | 0.02±0.007 | 0.19±0.048 |
| GD | 0.06±0.004 | 0.06±0.002 | 0.13±0.004 | 0.08±0.002 | 0.05±0.002 | 0.14±0.008 |
| TriTPP | 0.03±0.002 | 0.04±0.001 | 0.23±0.003 | 0.04±0.003 | 0.02±0.002 | 0.05±0.004 |
| ADD-THIN (Ours) | 0.02±0.004 | 0.02±0.002 | 0.19±0.013 | 0.03±0.006 | 0.02±0.001 | 0.10±0.030 |

Table 8: **Real-world data**: MMD ($\downarrow$) between the TPP distribution of sampled sequences and hold-out test set.

|  | PUBG | Reddit-C | Reddit-S | Taxi | Twitter | Yelp1 | Yelp2 |
|---|---|---|---|---|---|---|---|
| RNN | 0.04±0.005 | 0.01±0.002 | 0.02±0.003 | 0.04±0.001 | 0.03±0.003 | 0.07±0.005 | 0.03±0.001 |
| Transformer | 0.06±0.014 | 0.05±0.025 | 0.09±0.06 | 0.09±0.014 | 0.08±0.02 | 0.12±0.026 | 0.14±0.048 |
| GD | 0.11±0.023 | 0.03±0.001 | 0.03±0.001 | 0.1±0.002 | 0.15±0.011 | 0.12±0.01 | 0.1±0.001 |
| TriTPP | 0.06±0.001 | 0.09±0.002 | 0.12±0.003 | 0.07±0.007 | 0.04±0.002 | 0.06±0.005 | 0.06±0.004 |
| ADD-THIN (Ours) | 0.03±0.015 | 0.01±0.005 | 0.02±0.001 | 0.04±0.006 | 0.04±0.006 | 0.08±0.01 | 0.04±0.005 |

Table 9: **Synthetic data**: Wasserstein distance ($\downarrow$) between the distribution of the number of events of sampled sequences and hold-out test set.

|  | Hawkes1 | Hawkes2 | SC | IPP | RP | MRP |
|---|---|---|---|---|---|---|
| RNN | 0.03±0.007 | 0.01±0.002 | 0.00±0.003 | 0.02±0.006 | 0.02±0.002 | 0.01±0.004 |
| Transformer | 0.06±0.017 | 0.04±0.01 | 0.06±0.008 | 0.07±0.035 | 0.04±0.005 | 0.11±0.048 |
| GD | 0.16±0.016 | 0.13±0.012 | 0.5±0.025 | 0.42±0.009 | 0.28±0.039 | 0.5±0.035 |
| TriTPP | 0.03±0.003 | 0.03±0.001 | 0.01±0.0 | 0.01±0.001 | 0.02±0.003 | 0.03±0.001 |
| ADD-THIN (Ours) | 0.04±0.009 | 0.02±0.006 | 0.08±0.018 | 0.01±0.003 | 0.02±0.001 | 0.04±0.006 |

Table 10: **Real-world data**: Wasserstein distance ($\downarrow$) between the distribution of the number of events of sampled sequences and hold-out test set.

|  | PUBG | Reddit-C | Reddit-S | Taxi | Twitter | Yelp1 | Yelp2 |
|---|---|---|---|---|---|---|---|
| RNN | 0.02±0.004 | 0.01±0.004 | 0.05±0.013 | 0.02±0.002 | 0.01±0.001 | 0.04±0.004 | 0.02±0.002 |
| Transformer | 0.04±0.013 | 0.08±0.028 | 0.11±0.032 | 0.13±0.073 | 0.05±0.021 | 0.11±0.03 | 0.21±0.077 |
| GD | 0.54±0.054 | 0.02±0.004 | 0.16±0.013 | 0.33±0.007 | 0.07±0.062 | 0.26±0.012 | 0.25±0.007 |
| TriTPP | 0.03±0.003 | 0.09±0.001 | 0.09±0.001 | 0.04±0.001 | 0.01±0.001 | 0.03±0.006 | 0.04±0.002 |
| ADD-THIN (Ours) | 0.02±0.009 | 0.03±0.007 | 0.04±0.002 | 0.03±0.007 | 0.01±0.004 | 0.04±0.006 | 0.02±0.006 |

### E.2 Forecasting results with standard deviations

Table 11: Wasserstein distance ($\downarrow$) between forecasted event sequence and ground truth reported for 50 random forecast windows on the test set.

|                    | PUBG        | Reddit-C     | Reddit-S     | Taxi        | Twitter     | Yelp1       | Yelp2       |
|--------------------|-------------|--------------|--------------|-------------|-------------|-------------|-------------|
| Average Seq. Length | 76.5        | 295.7        | 1129.0       | 98.4        | 14.9        | 30.5        | 55.2        |
| RNN                | 6.15±2.53   | 35.22±4.02   | 39.23±2.06   | 4.14±0.25   | 2.04±0.08   | 1.28±0.03   | 2.21±0.06   |
| Transformer        | 2.45±0.21   | 38.77±10.68  | 27.52±5.24   | 3.12±0.1    | 2.09±0.07   | 1.29±0.1    | 2.64±0.24   |
| GD                 | 5.44±0.2    | 44.72±1.77   | 64.25±4.45   | 4.32±0.3    | 2.16±0.23   | 1.52±0.15   | 4.25±0.46   |
| ADD-THIN (Ours)    | 2.03±0.01   | 17.18±1.18   | 21.32±0.42   | 2.42±0.03   | 1.48±0.03   | 1.0±0.02    | 1.54±0.04   |

Table 12: Count MAPE $\times 100\%$ ($\downarrow$) between forecasted event sequences and ground truth reported for 50 random forecast windows on the test set.

|                    | PUBG         | Reddit-C     | Reddit-S     | Taxi        | Twitter     | Yelp1       | Yelp2       |
|--------------------|--------------|--------------|--------------|-------------|-------------|-------------|-------------|
| Average Seq. Length | 76.5         | 295.7        | 1129.0       | 98.4        | 14.9        | 30.5        | 55.2        |
| RNN                | 1.72±0.65    | 5.47±0.92    | 0.68±0.07    | 0.54±0.02   | 0.95±0.08   | 0.59±0.02   | 0.72±0.03   |
| Transformer        | 0.65±0.11    | 7.38±2.51    | 0.55±0.14    | 0.46±0.04   | 1.18±0.09   | 0.63±0.08   | 0.99±0.11   |
| GD                 | 1.66±0.06    | 10.49±0.42   | 1.33±0.12    | 0.71±0.05   | 1.43±0.2    | 0.78±0.1    | 1.65±0.2    |
| ADD-THIN (Ours)    | 0.45±0.005   | 1.07±0.19    | 0.38±0.02    | 0.37±0.02   | 0.69±0.03   | 0.45±0.02   | 0.5±0.03    |

### E.3 Sampling runtimes

We compare sampling runtimes on an NVIDIA GTX 1080 Ti across the different models in Figure 5. ADD-THIN maintains near-constant runtimes by refining the entire sequence in parallel. The autoregressive baselines *RNN* and *Transformer* show increasing runtimes, with longer sequences surpassing ADD-THIN's runtime. *TriTPP* is a highly optimized baseline computing the autoregressive interactions between event times in parallel by leveraging triangular maps, resulting in the fastest runtimes. Lastly, *GD* is autoregressive in event time and gradually refines each event time over 100 diffusion steps, leading to the worst runtimes.

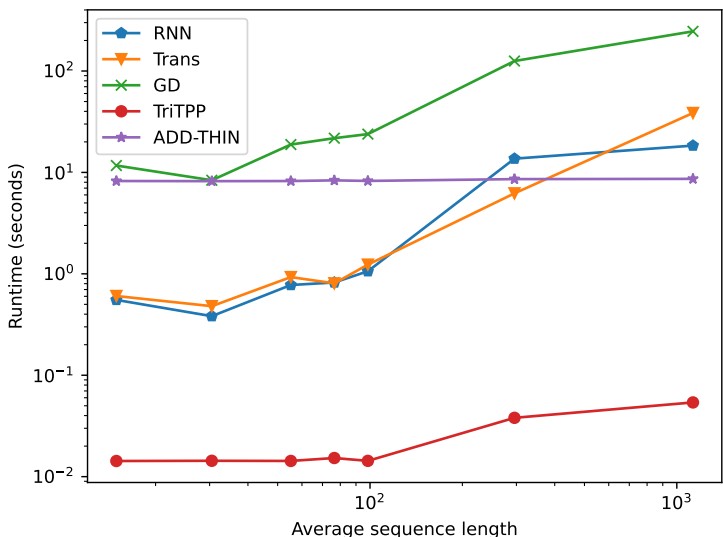

Figure 5: Sampling runtime for a batch of 100 event sequences averaged over 100 runs. We report the trained model's sampling times for the real-world datasets with different sequence lengths (from left to right: Twitter, Yelp 1, Yelp 2, PUBG, Taxi, Reddit-C, Reddit-A).

