# OpenReview forum: "Add and Thin: Diffusion for Temporal Point Processes"
_NeurIPS.cc/2023/Conference — NeurIPS 2023 poster_

### Official Review · Reviewer_6sim · 2023-06-30

**Soundness:** 3 good
**Presentation:** 3 good
**Contribution:** 3 good
**Rating:** 7
**Confidence:** 4

**Summary:**

The submission derive a probabilistic diffusion model for TPPs. By proposing the Add-Thin framework, the proposed method can naturally handles the continuous and discrete nature of point processes and directly models the whole event sequences. Compared with the traditional autoregressive approaches for the temporal point process (TPP), the proposed framework is immune to the accumulation of errors caused by sequential sampling.

**Strengths:**

The submission proposed a framework that learns the probabilistic mapping from complete noise, i.e., a homogeneous Poisson process (HPP), to data. More specifically, a model is learned to reverse the noising process of adding (superposition) and removing (thinning) points from the event sequence by matching the conditional inhomogeneous denoising intensity. This idea is very interesting.

**Weaknesses:**

I do not find any obvious weaknesses.

**Questions:**

None.

**Limitations:**

The authors discussed the limitations and potential negative societal impact of their work.

---

> ### Author Rebuttal · Authors · 2023-08-09
>
> Thank you for your review and appreciation of our work.

---

> > ### Comment · Reviewer_6sim · 2023-08-15
> > **Thank you for your response**
> >
> > Thank you for your response. I think this is an interesting work and keep my original score.

---

### Official Review · Reviewer_RwFj · 2023-07-05

**Soundness:** 3 good
**Presentation:** 1 poor
**Contribution:** 3 good
**Rating:** 6
**Confidence:** 4

**Summary:**

The forward process adds points from homogeneous poisson process (HPP) into the sequence and removes points from original sequence ($\mathbf{t}^{(0)}$). The goal is such that $\mathbf{t}^{(n)}$ is HPP. The neural network is trained to approximate missing information about $\mathbf{t}^{(0)}$, e.g. figure out which points from HPP came from $\mathbf{t}^{(0)}$ and which points were removed at intermediate steps. The thinning and superposition theorems were used multiple times to derive intensities of various sets, where the sets categorize the points.

**Strengths:**

1. The adding and thinning approach, training via classification and matching, to modeling and generating temporal point processes is original and interesting.

**Weaknesses:**

1. The writing and notation is extremely confusing. I read the paper multiple times.

1a. e.g. please clearly specify the observation, is it a time sequence $x(t), t\in[0,T]$ where $x(t) = 0$ for no event, $x(t) = 1$ for event, discretized in time. or is it a varying length sequence $\mathbf{t}=(t_1,...,t_K), 0< t_1<...<t_K\leq T$ with $K$ arrival times. also, section 3.3 writes about sequence embedding to get an input in $R^d$ but I am still confused.

1b. e.g. what are the sets A and F? what are the differences between B and C, D and E? B seems to be the points preserved from $\mathbf{t}^{(0)}$ all the way to HPP, C seems to be the points preserved from $\mathbf{t}^{(0)}$ but removed in the next step. is there a clearer way of writing the material in section 3.2?

1c. perhaps the background section can be shortened if we need more space to explain the method clearly.

1d. would potentially increase the score if the model and method is clearly explained.

**Questions:**

1. Would like to introduce this recent other work that connects point processes and diffusions. in this case, the interarrival times of a point process is related to the length of a diffusion excursion. https://openreview.net/forum?id=KIlwyX7nCi

2. What is the benefit of applying the proposed model compared to other methods of generating sequences?

**Limitations:**

1. Please address the potential limitations of using a neural network to approximate missing information about $\mathbf{t}^{(0)}$.

---

> ### Author Rebuttal · Authors · 2023-08-09
>
> Thank you for your comments and feedback. In the following, we address the specific concerns and questions.
>
> > W.1a. Obervations
>
> Generally a realization of a TPP can be represented as a sequence of strictly increasing arrival times: $\mathbf{t} = (t_1, \dots, t_K)$,  $0 < t_1 < \dots < t_K \leq T$. Equivalently, a realization can also be represented by its counting measure $N(t) = \smash{\sum_i^K \mathbb{I}(t_i \le t)}$, for $t \in [0, T]$. In the implementation of our method we leverage the arrival times $\mathbf{t}$, which we will make more explicit by adding the formal definition to the introduction of $\mathbf{t}^{(0)}$ in line 101. Furthermore, we employ a temporal embedding followed by a sequence embedding with a CNN to attain embeddings of $\mathbf{t}^{(n)}$ and subsequently parameterize the posterior.
> > W.1b. Unclear distinction between sets A,...,F
>
> We have revised section 3.2 and present an improved explanation of the sets in the response to all reviewers to address this weakness.
>
> > W.1c. Shorten background section if more space is needed
>
> Thank you for the suggestion. Given the more concise revision of section 3.2 and the additional page for the camera-ready version we will not need to cut down on the background section. Additionally, we will use the additional page to describe the method in more detail and explain the motivation behind the design choices.
>
> > W.1d. I would potentially increase the score if the model and method is clearly explained.
>
> We hope that the revision of section 3.2 in the response to all reviewers presents a clearer explanation of the model and method. Further, we would like to add the following revisions and points to better explain the model and method:
>
> **Section 2.2**
>
> [Line 85-91]
>
> Diffusion models [16, 40] are a class of latent variable models that learn a generative model to reverse a fixed probabilistic noising process $x_{0} \to x_{1} \to \dots \to x_N$, which gradually adds noise to clean data $x_0$ until no information remains, i.e., $x_{N}\sim p(x_N)$. For continuous data, the forward process is usually defined as a fixed Markov chain $q(x_{n} \mid x_{n-1})$ with Gaussian transitions. Then the Markov chain of the reverse process is captured by approximating the true posterior $q(x_{n-1}|x_{n}, x_{0})$ with a model $p_{\theta}(x_{n-1}|x_{n})$. Ultimately, sampling new realizations $x_0$ from the modeled data distribution $p_{\theta}(x_{0}) = \int p(x_{N}) \prod^N_{n=1} p_{\theta}(x_{n-1}| x_{n}) dx_{1},\cdots,x_{N}$ is performed by starting with a sample from pure noise $x_{N}\sim p(x_N)$ and gradually denoising it with the learned model over $N$ steps $x_{N} \to x_{N-1} \to \dots \to x_0$.
>
> **Section 3.3**
>
> [Line 173-177]
>
> In the previous section we have derived the intensity $\lambda_{n-1}(t | \mathbf{t}^{(0)}, \mathbf{t}^{(n)})$ of the posterior $q(\mathbf{t}^{(n-1)}|\mathbf{t}^{(0)}, \mathbf{t}^{(n)})$ for the reverse process, i.e., the intensity of points at step $n-1$ given $\mathbf{t}^{(0)}$ and $\mathbf{t}^{(n)}$. Now we want to approximate this posterior using a model $p_{\theta}(\mathbf{t}^{(n-1)} | \mathbf{t}^{(n)}, n)$ to learn to sample points $\mathbf{t}^{(n-1)}$ given only $\mathbf{t}^{(n)}$. As we are only missing information about $\mathbf{t}^{(0)}$ we will learn to model $\lambda^{(B)}(t)$ and $\lambda^{(A\cup C)}(t)$ to approximate $\hat{\mathbf{t}}^{(0)} \approx \mathbf{t}^{(0)}$ (see Figure 3.) for each $n$ and then have access to the full posterior intensity to reverse the noising process.
>
> [Line 178]
>
> To condition our model on $\mathbf{t}^{(n)}$ and $n$ we propose the following embeddings. [...]
>
> > Q1. Connection to work on diffusion excursion for TPPs.
>
> Thank you for introducing us to this exciting concurrent work. In this paper, they model TPPs by relating the continuous latent state in a diffusion process to the event times by modeling the interevent times as the length of a diffusion excursion. Thereby, the proposed method is still autoregressive and more strongly related to TPP models, which model event times conditional on a continuous latent process (see, e.g. [4, 20, 11]). Nevertheless, the proposed connection between the continuous stochastic processes and TPPs is very interesting, and we will make sure to reference and discuss this work in our related work section.
>
> > Q2. Benefit of Add-Thin compared to other methods
>
> Our method is the first diffusion-based TPP model that generates entire event sequences, thereby overcoming some of the shortcomings of autoregressive models in forecasting. Additionally, by replacing the autoregressive parametrization in event time with an gradual refinement process of the entire event sequence for a fixed number of N diffusion steps, our model is better suited to model very long event sequences (c.f. sampling speed in response to all reviewers). Furthermore, the iterative refinement of the event sequence allows us to leverage simple and shared layers to model the long-range interaction between events, a task for which attention (computational complexity) and RNN-based encoders are known to struggle.
>
> > Please address the potential limitations of using a neural network to approximate missing information about $\mathbf{t}^{(0)}$.
>
> Our model is trained to generate samples $\mathbf{t} \sim q(\mathbf{t}^{(0)})$, by optimizing the ELBO of our probabilistic model (see discussion of the relationship of our loss and the ELBO in the response to reviewer 1d9q). In that context, the applied neural networks to approximate the posterior are universal function approximators and can in theory model the posterior distribution arbitrarily well.

---

> > ### Comment · Reviewer_RwFj · 2023-08-15
> >
> > W.1a. Obervations
> > - Please further clarify  ``temporal embedding followed by a sequence embedding''. In addition, is it e.g. $\mathbf{t}^{1}=(1,5,7,9)$, $\mathbf{t}^{2}=(2,5,6)$, where $\mathbf{t}^{1}$ and $\mathbf{t}^{2}$ are two data samples, and the lengths may be different? What is the input to the network at each stage? Or is it possible to make code available, thanks!
> >
> > W.1b. Unclear distinction between sets A,...,F.
> > - Reviewer 1d9q had the same comment too. In the initial submission, I was guessing what the model and method is. Read the response to all reviewers. Is it also possible to have a figure, thanks!
> >
> > Q1. Connection to work on diffusion excursion for TPPs.
> > - Thank you for pointing me at those references!
> >
> > Please address the potential limitations of using a neural network to approximate missing information about $\mathbf{t}^{(0)}$
> > - Read the response to reviewer 1d9q. Approximating Dirac deltas with Gaussians may not work as well...
> >
> > Read all the reviews and the responses, I am positive and interested in the work, increasing score to 5 for now, will wait for the authors to respond.

---

> > > ### Author Response · Authors · 2023-08-17
> > >
> > > Thank you for your response and active engagement in the rebuttal discussion. We are delighted that you are positive and interested in our work. In the following, we would like to clarify your comments:
> > >
> > > ### Observations
> > > > In addition, is it, e.g., $\mathbf{t}^{1} = (1,5,7,9)$, $\mathbf{t}^{2} = (2,5,6)$, where $\mathbf{t}^{1}$ and $\mathbf{t}^{2}$ are two data samples, and the lengths may be different?
> > >
> > > Indeed, instances or data samples of a TPP can have different lengths and could, for example, look like $(1,5,7,9)$ and $(2,5,6)$. In other words, both the arrival times and number of points are random. This defining property of a TPP is introduced in the background section 2.1 lines [59-61]. Further, we would like to highlight the notational difference in your response. With the superscript $\mathbf{t}^{(n)}$, *we* refer to an event sequence at the $n$-th diffusion step, where $\mathbf{t}^{(n)} \sim \lambda_n$ and not to different samples from the data distribution.
> > > > Clarify ``temporal embedding followed by a sequence embedding'' and the input to the network at each stage (can you maybe add code).
> > >
> > > The temporal and sequence embedding were explained in Sec. 3.3 lines [178-183] with the additional sentence in our last response and further depicted in Fig. 3. Here, we will additionally present some background and slightly informal explanation of the embedding types for TPPs:
> > >
> > > **Temporal (Positional) embedding:** Applying temporal embeddings, e.g., log-transform and trigonometric functions (similar to the positional embeddings of Transformer [3]), to event times is a common practice for TPPs (see Sec. 3.1 of [1]). With a temporal embedding, we refer to a function $\mathbb{R}_{>0} \to \mathbb{R}^d$, that maps a time, e.g., $t_i \in \mathbf{t}$ independently to a $d$-dimensional embedding space. Then for each  ${t}_i$, we have a feature vector $\mathbf{c}_i$ encoding its temporal information.
> > >
> > > **Sequence embedding:** To attain an event embedding (also sometimes called context embedding) $\mathbf{e}_i$ for each event $i$ that also incorporates information about all other events, we apply a three-layered CNN with dilation and circular padding on the sequence of temporally embedded event times $(\mathbf{c}_1, \cdots, \mathbf{c}_K)$. Ultimately, a global sequence embedding $\bar{\mathbf{e}}$ is attained by computing the average over $(\mathbf{e}_1, \cdots, \mathbf{e}_K)$.
> > >
> > > The code will be published upon acceptance, but we present a simplified pseudo-code of our implementation of the sequence embedding here:
> > >
> > > ```
> > > # temporal embedding
> > > time_embedding = self.time_encoder([x_n.time, x_n.tau]) # Kxd
> > > # event embedding
> > > event_embedding = self.cnn(time_embedding) # Kxd
> > > # sequence embedding
> > > seq_embedding = event_embedding.mean(dim=0) # d
> > > ```
> > >
> > > As described in Sec. 3.3 line [185; 196] and Fig. 3, ```time_embedding``` $\mathbf{c}_i$ and ```event_embedding``` $\mathbf{e}_i$ are the inputs to the classifier and ```seq_embedding``` $\bar{\mathbf{e}}$ is the input to the MLPs parameterizing the intensity function.
> > >
> > > ### Set distinction
> > > > Is it also possible to have a figure (Distinction between sets A,...,F)?
> > >
> > > We might be wrong but believe that Fig. 1 (right) and 2 (left) are what you are looking for. Fig. 2 presents the case distinction of the sets **A**,...,**F** with a Venn diagram as the logical relations between $\mathbf{t}^{(0)}$, $\mathbf{t}^{(n-1)}$ and $\mathbf{t}^{(n)}$. Further, in Fig. 1, the denoising process illustrates the case distinction for one specific example of $\mathbf{t}^{(0)}$ and $\mathbf{t}^{(n)}$. Please note that the illustrated posterior intensity can be separated into the sets **B-E**: The base intensity relates to $\lambda^{(E)}(t)$, while the highest peaks (Diracs) refer to $\lambda^{(B)}(t)$, i.e., the points that are in $\mathbf{t}^{(0)}$ and $\mathbf{t}^{(n)}$. The medium peaks and lower peaks (thinned Diracs) represent $\lambda^{(D)}(t)$ and $\lambda^{(C)}(t)$, respectively. Please let us know if this addresses your point, and we will, for the camera-ready version, add an additional reference to the denoising process in Fig. 1 in the discussion of the reverse process in Sec. 3.2.
> > >
> > > ### Approximation
> > >  > Approximating Dirac deltas with Gaussians may not work as well...
> > >
> > > A Gaussian is the standard approximation of the Dirac delta function and can, in the limit $\sigma \to 0$, perfectly approximate it [2, 4]; a property that is not true for many approximations of other TPP models (e.g., parametric distribution for the intensity function of *RNN*, *Trans* and *TriTPP*). Furthermore, the KL divergence between Dirac and our Gaussian is directly minimized to ensure the best possible approximation.
> > >
> > > ### References
> > > 1. Citation 38 in the paper
> > > 2. Ghatak, Ajoy, et al. "The dirac delta function." Quantum Mechanics: Theory and Applications (2004)
> > > 3. Citation 42 in the paper
> > > 4. Saichev et al. "Chapter 1: Basic definitions and operations", Distributions in the Physical and Engineering Sciences (1997)

---

> > > > ### Comment · Reviewer_RwFj · 2023-08-17
> > > >
> > > > Thanks for the clarification! Increased score to 6.
> > > >
> > > > Q. additional reference to the denoising process in Fig. 1 in the discussion of the reverse process in Sec. 3.2.
> > > > Yes please, for the camera-ready, it would be very helpful to label each peak/ addition/ thinning with the set.

---

### Official Review · Reviewer_1xQ2 · 2023-07-06

**Soundness:** 2 fair
**Presentation:** 1 poor
**Contribution:** 3 good
**Rating:** 5
**Confidence:** 4

**Summary:**

The paper proposes a probabilistic diffusion model for TPPs, ADD-THIN, that naturally handles the continuous and discrete nature of point processes and directly models whole event sequences. While autoregressive methods are expressive in modeling event sequences, ADD-THIN does not suffer from the accumulation of errors caused by sequential sampling for long-term forecasting applications. The proposed method matches the performance of state-of-the-art models in density estimation and outperform them for forecasting on both synthetic and real-world datasets.


**Strengths:**

- To the best of my knowledge, this work represents the first attempt to combine diffusion models and Temporal Point Processes. The potential of generating complete sequences without relying on auto-regressive mechanisms presents a promising avenue for future research. It is worth noting that this paper shares similarities with the unpublished work available at: https://slideslive.com/embed/presentation/38922857.

- The proposed method is compared to multiple baselines for both unconditional and conditional sampling (forecasting).


**Weaknesses:**


- The paper is poorly written and hard to read.

- There are multiple statements that are either inaccurate or without references
	- "The intensity of a TPP describes the expected number of points"
	- Line 65-70, you either need to prove it or provide references that do so.
	- Line 72, "the expected number of points on [0, T] follows a Poisson distribution with rate ..."
	- "Even though Poisson processes can model seasonality"
		- How does a Poisson model seasonality?
	- "We know that conditioned on ... is a random sequence of events with a density that can be completely described by its (unconditional) intensity function"
	- "Since the sets are disjoint, ..."
	- Is there a difference between a counting process and measure? an intensity function and measure? You use these terms interchangeably.


- Multiple TPP datasets that are used in the experiments have been shown to be inappropriate benchmarks for neural TPPs. See https://openreview.net/forum?id=3OSISBQPrM

- Overall, the presentation of the paper could be greatly improved, especially Section 3. Specifically, it feels like adding more formal notations (e.g. for Section 3.2.) would allow to shorten the text, while helping the reader more easily grasp the idea behind the method. For instance, why defer to the Appendix the pseudo-code for the sampling procedure?

Additionally, several design choices are not clearly motivated, while the experiments and evaluation of the proposed model falls a bit short of expectations. These concerns are summarized in the points below:

1) Why include both $\boldsymbol{e}i$ and $\boldsymbol{c}_i$ in the parametrization of $g\theta$? Are the $\boldsymbol{c}_i$ constructed from neighboring events? Otherwise, it looks like there is redundancy between the information contained in the two embeddings.

2) There is no real motivation behind the use of a positional encoding for $n$. Why do you need it?

3) What is the motivation for using an unormalized mixture of Gaussians to parametrize $\lambda^{(A \cup C)}$? Have you considered alternative parametrizations that can be found in the neural TPP literature, e.g. an unmarked version of RMTPP [1]? Additionally, more details regarding the architectures employed would be appreciated (e.g. what is $\sigma$, and what is the model behind the MLP on page 5?).

4) In the NLL objective of equation 5, is the objective solely computed on times from $A \cup C$? If yes, this should be indicated explicitly in the objective.

5) On page 6, it is not clear how you find that $\Lambda^{(A \cup C)}(T) = K \sum_{i}^H w_i$. Given your definition of $\lambda^{(A \cup C)}(t)$, I would have expected this result to hold only if the integral was taken over $-\infty, \infty$.

6) The paragraph on 'conditional sampling' requires more detailed explanations, as it is impossible to see from the text how you handle this scenario. Specifically, how is $\boldsymbol{h}$ integrated into $g_{\theta}$ and $\lambda^{(A \cup C)}(t)$?

6) As $\hat{t}^{(0)}$ is successively approximated at each time step $n$ during the reverse process, how can you ensure that the propagation of errors is mitigated and won't affect significantly the generated sequences after $N$ steps?

7) Why only reporting the MAPE between the generated and truth sequences lengths? Isn't there a way to evaluate the quality of the generated samples within a sequence itself (e.g. using a measure of distance)?

[1] Du, Nan and Dai, Hanjun and Trivedi, Rakshit and Upadhyay, Utkarsh and Gomez-Rodriguez, Manuel and Song, Le. (2016). Recurrent Marked Temporal Point Processes: Embedding Event History to Vector. SIGKDD.



**Questions:**

See Weaknesses.

**Limitations:**

The authors have a section on some limitations of their work, although a more in-depth discussion would have been appreciated. Specifically, they mention that sampling from their diffusion model can be quite expensive, but they do not provide computional trade-offs in terms of time/resources across baselines.

---

> ### Author Rebuttal · Authors · 2023-08-09
>
> Thank you for the extensive review, feedback, and questions. In the following, we address the raised concerns and questions.
> > W.1,4 Readability and understandability (3, 3.2)
>
> Considering the character limit, we refer to both our response to all reviewers, where we outline suggested enhancements for the reverse process and our answer to reviewer RwFj's comments regarding the method description. Further, we will use the additional page of the camera-ready version to move the pseudo-code of the sampling procedure from the App. to Sec. 3.4. Lastly, subsequent segments in this rebuttal are dedicated to address questions and misunderstandings in our method description.
> > W.2 Clarifications and references
> 1. "The intensity is defined as:"; line 64 "The intensity can be interpreted as the expected number of events per unit of time."
> 2. Reference to [7]
> 3. "the number of points..."
> 4. "...inhomogeneous Poisson processes..." The time-varying intensity of an IPP can, e.g., capture the fact that more events happen at a certain day of the week, known as seasonality.
> 5. Rewritten as part of the response to all reviewers.
> 6. "Since the sets B-E..."
> 7. As described in Sec. 2.1, a counting process (in our case TPP) can be described by its intensity measure or less technical function, while a realization can be represented by its counting measure. We will use "intensity function" throughout to make the text more consistent.
>
> > W.3 Benchmark datasets
>
> Thank you for pointing out this relevant concurrent work, raising concerns in the evaluation of autoregressive TPP models with teacher forcing for some widely used TPP benchmark datasets. We were unable to incorporate its findings since the work was published 2 months after the NeurIPS 2023 submission deadline. In general, we agree that this is an important concern. However, likely these findings are not directly applicable to the setting considered in our work as we consider tasks, metrics, and model architectures that are all different and have found significant empirical differences between the baselines even on these datasets. We will investigate this issue in more detail and add a discussion to the revised version of the paper.
> > Design choices:
>
> We clarify the motivation for the design choices in the following:
> 1. In theory, $e_i$ can contain all information of $c_i$, but we have found that providing the event time explicitly gives better results.
> 2. Positional encodings for n were first proposed in [16] to improve the model capacity. To our knowledge, all current diffusion models leverage positional encodings as $n$ strongly affects the posterior.
> 3. In our diffusion model, all posterior intensities are inhomogeneous PP (c.f., PP section of the parametrization of RMTPP). The Gaussian distribution ($\sigma\rightarrow0$) is a standard approximation of the Dirac delta function in the posterior (see also answer to reviewer 1d9q). We explored other options, such as mixture of Beta and Weibull, but found their parametric form to restrict the modeling of Dirac delta functions. The MLPs have two layers, a hidden dimension of 32 and a ReLU activation (we add additional information to the App.). Lastly, $\sigma$ refers to the sigmoid activation, which we will make explicit.
> 4. The objective is only computed for $A\cup C$ as indicated in Eq. 5. Note that the sum in Eq. 5 could equally be written as $\sum_{t_i\in\mathbf{t}^{(A\cup C)}}$.
> 5. We employ re-normalized truncated Gaussians on the interval [0,T]. We will add this information to the paragraph.
> 6. The intensity and classifier get the positional embedding of the diffusion time n as an input. The history embedding $\mathbf{h}$ is simply added to this embedding. We will revise the last sentence to even better convey this information.
> 7. When sampling from any diffusion model, a noisy sample is gradually refined to produce a sample from the learned data distribution. Hence, each subsequent diffusion step can correct for errors (or rather noise) of the previous ones. In our case, we refine the **entire event sequence** at every diffusion step to finally produce a data sample. This starkly contrasts TPP models that are autoregressive in the event time and do not adjust errors in earlier events. This difference becomes especially evident in our conditional sampling experiment, where the accumulation of error in event time restricts the forecast capability of autoregressive TPP models. Lastly, we would like to use this chance to refer to our response to reviewer 1d9q, where we elaborate on the relation of our loss and the ELBO and training our model by minimizing the KL divergence between our approximate and the true posterior.
> 8. We think there has been a slight misunderstanding. We do not only report the MAPE for the sequence lengths but also report the Wasserstein distance between the generated and ground truth sequence. This Wasserstein distance measures the distance between the two counting measures representing each event sequence.
>
> > Computational complexity
>
> In our response to all reviewers, we added a plot to the PDF that compares the sampling runtimes for the different models. We find that for longer sequences, the runtime of our unoptimized model is comparable to autoregressive models. This interesting finding can be explained on a conceptual level as our diffusion model presents a different trade-off: Instead of the autoregressive nature of common TPP models in event time, our model in parallel refines the entire sequence for a fixed number of N diffusion steps. Thereby, the sampling is almost constant across the different sequence lengths.
> > Similarity to unpublished work
>
> Thank you for pointing out this work, which discusses the general idea of a hierarchical composition of simple intensity functions and defining invertible probabilistic transformation (destructors) of point processes. If you could provide us with a preprint or reference to this work, we would be happy to discuss and reference it.

---

> > ### Comment · Reviewer_1xQ2 · 2023-08-15
> > **Rebuttal**
> >
> > Thank you for your clarifications on my concerns regarding the design choices, and for taking my recommendations into account. There was indeed a misunderstanding regarding the metrics employed. Upon reading your updated description of the reverse process and your replies to other reveiwers, I have increased my score. Nevertheless, in light of the fact that we won't have the opportunity to review the revised paper, I remain hesitant to recommend its acceptance.

---

> > > ### Comment · Reviewer_RwFj · 2023-08-15
> > >
> > > The unpublished work is ``Deep Point Process Destructors'' David I. Inouye, NeurIPS Workshop on Learning with Temporal Point Processes, 2019.

---

### Official Review · Reviewer_1d9q · 2023-07-17

**Soundness:** 4 excellent
**Presentation:** 3 good
**Contribution:** 4 excellent
**Rating:** 7
**Confidence:** 3

**Summary:**

The paper introduces a novel probabilistic diffusion framework for temporal point process. Its significance lies modeling a whole event sequence directly, overcoming common limitations of autoregressive models.

**Strengths:**

Originality:  This paper is very novel. It connects diffusion models with TPPs and model a whole event sequence , overcoming issues from autoregressive models.

Quality: The formulation of the approach and method itself are sound by leveraging two properties of point process: thinning and superpositions. The discussion of the reverse process are reasonable by discussing 4 cases of transition from nth step to n-1 step.  The empirical results demonstrate the good performance of the proposed approach, esp. in forecasting tasks.

Clarity:  The paper is overall well-presented. Improvement can be made by better explaining A,B,C,D,E,F sets.

Significance:  The paper introduces a new way to model event sequences probabilistically  which is appealing and significant to the TPP community.


**Weaknesses:**

A.	Experiments. The experimental evaluation on density estimation. The results of Add and Thin is only comparable to Intensity-free model by Shchur (RNN in the paper), if not worse.
B.	Application and Limitation. One minor limitation is it only deals with TPP without marks. It would be interested in see something for Marked TPP.
C.	Overall presentation. It took me a while to understanding the notation of sets A-F. It would be nice to include such information in the caption of figure 2.
D. No codes are given, I am not sure how reproducible the results are.


**Questions:**

A.	Posterior approximation.  How good is the posterior approximation in Add and Thin? Are there any theoretical insights.
B.	Neural network. Could the authors provide some movitation of using 3-layered-d CNN?
C.	Loss. 1). Is the mixture of Gaussians eqn. (4) used to model the conditional intensity good for modeling an event time(or interevent time) t which is positive? Add and Thin is a latent variable model with latent variables t^{n-1}’s. Usually int the diffusion models reference 40 in the paper, latent variable models leverage variational inference to maximize the evidence lower bounds. But this model is not. Could the author shed some insights on this?
D.	Cox process models. Do the authors have any experiment results with these baselines form reference [5,8,17]?
E.	Experiments. How many diffusion steps (n) are used?


**Limitations:**

The limit of the paper according to the authors is it is limited to TPP without marks. In addition, current sampling routines may not be efficient. TPP models can be applied applied to many fields, where common application domains  include traffic, social networks, and electronic health records.

---

> ### Author Rebuttal · Authors · 2023-08-09
>
> Thank you for your thorough review, appreciation of our work, and suggestions to further improve our paper.
> > W.A Comparison to Shchur et al.
>
> We want to point out that the intensity-free model by Shchur et al. is a very strong baseline, and both our model and this baseline achieve near-perfect metrics in the density estimation task on most datasets. Further, the proposed model is primarily motivated through the conditional sampling/forecasting task, where our model outperforms all baselines.
> > W.B Minor point: Marks
>
> We agree that including marks would pose an interesting extension of our model. In this paper, we focused on deriving a sound diffusion process for TPPs to tackle the complex task of accurately modeling event sequences in time and leave the extension to marks for future work.
> > W.C Presentation of sets
>
> Please refer to our response to all reviewers for an improved presentation of the sets and reverse process.
> > W.D Reproducibility of results
>
> We will release a well-documented version of our code upon acceptance.
> > Q. Relation of the loss to the ELBO
>
> Thank you for this important remark. Our model's objective function is indeed equivalent to the ELBO of a diffusion model, as shown below. We will add this discussion to the appendix and reference it in Sec. 3.3.
>
> The ELBO of diffusion models [2] is derived as follows:
> \begin{equation}
> \begin{split}
>     \mathcal{L}\_{ELBO} = \mathbb{E}\_{q} \Big[\underbrace{D\_{KL}\left(q(\mathbf{t}^{(N)}|\mathbf{t}^{(0)})\parallel p(\mathbf{t}^{(N)})\right)}\_{\mathcal{L}\_{N}}+\sum_{n=2}^{N} \underbrace{D\_{KL}\left(q(\mathbf{t}^{(n-1)}|\mathbf{t}^{(0)},\mathbf{t}^{(n)})\parallel p_{\theta}(\mathbf{t}^{(n-1)}|\mathbf{t}^{(n)})\right)}\_{\mathcal{L}\_{n}}-\underbrace{\log p_{\theta}(\mathbf{t}^{(0)}|\mathbf{t}^{(1)})}\_{\mathcal{L}\_{0}}\Big].
> \end{split}
> \end{equation}
>
> The Janossi density [3] of a TPP allows us to represent each element of the ELBO in terms of our derived inhomogeneous (approximate) posterior intensities (Sec. 3.2.) as follows:
>
> \begin{equation}
>     p(\mathbf{t}) =e^{-\int_0^T \lambda(t)dt} \prod_{t_i\in \mathbf{t}} \lambda(t_i).
> \end{equation}
>
> $\boldsymbol{\mathcal{L}\_{N}}$: $\mathcal{L}\_{N}$ is constant as $q(\mathbf{t}^{(N)}|\mathbf{t}^{(0)})$ and $p(\mathbf{t}^{(N)})$ have no learnable parameters.
>
> $\boldsymbol{\mathcal{L}\_{0}}$: We directly train our model to optimize this likelihood term as described in Sec. 3.3.
>
> $\boldsymbol{\mathcal{L}\_{n}}$: The KL divergence between two densities is defined as:
> \begin{align}
>     D\_{KL}(q\parallel p_{\theta}) = \mathbb{E}\_{q}\left[\log (q(\mathbf{t}^{(n-1)}|\mathbf{t}^{(0)},\mathbf{t}^{(n)}))-\log (p_{\theta}(\mathbf{t}^{(n-1)}|\mathbf{t}^{(n)}))\right],
> \end{align}
> where only the right-hand side relies on $\theta$, letting us minimize the KL divergence by maximizing the expectation over $\log(p_{\theta}(\mathbf{t}^{(n-1)}|\mathbf{t}^{(n)}))$.
>
> To add some additional context to the KL divergence in $\mathcal{L}\_{n}$ and similar to the derivation of the posterior, we will further distinguish three cases:
>
> - **Set E:** Our parametrization uses the same intensity function for $q(\mathbf{t}^{(n-1)}|\mathbf{t}^{(0)},\mathbf{t}^{(n)})$ and $p_\theta(\mathbf{t}^{(n-1)}|\mathbf{t}^{(n)})$, which does not rely on any learned parameters.
> - **Set B \& D:** $q(\mathbf{t}^{(n-1)}|\mathbf{t}^{(0)}, \mathbf{t}^{(n)})$ and $p_\theta(\mathbf{t}^{(n-1)}|\mathbf{t}^{(n)})$ are defined by Bernoulli distributions over each element of $\mathbf{t}^{(n)}$. The BCE minimizes the KL divergence between the two distributions as proposed in Sec. 3.3.
> - **Set C**: Minimizing this KL divergence by maximizing the $\mathbb{E}\_{q}\left[\log(p_{\theta}(\mathbf{t}^{(n-1)} |\mathbf{t}^{(n)}))\right]$ is proposed in Sec. 3.3 to learn $\lambda_{\theta}^{(A\cup C)}(t)$. Note that $q(\mathbf{t}^{(n-1)}|\mathbf{t}^{(0)},\mathbf{t}^{(n)})$ is sampled from by independently thinning $\mathbf{t}^{(0)}\setminus\mathbf{t}^{(n)}$. Consequently, by minimizing the NLL of our intensity $\lambda_{\theta}^{(A\cup C)}(t)$ with regards to $\mathbf{t}^{(0)}\setminus\mathbf{t}^{(n)}$, we optimize the expectation in closed form.
>
> > Q.A Posterior approximation
>
> Considering our ELBO answer, we directly minimize the KL divergence to match the true posterior. Here, the posterior intensity $\lambda^{(A\cup C)}$ consists of Dirac delta functions, which in the limit $\sigma\rightarrow0$ can be perfectly approximated by Gaussians [1]. Further, we use a universal function approximator (MLP) to parameterize the Bernoulli distribution in the classification.
> > Q.B Movitation CNN
>
> We use a CNN with dilation and circular padding as it's computationally much more efficient than attention based-encoders and captures long-range dependencies better than RNN-based ones. We'll clarify this choice in Sec. 3.3 in the final version.
> > Q.C Gaussian mixture
>
> We use truncated Gaussians, ensuring positive intensity only on [0, T].
> > Q.D Cox processes
>
> Cox processes are traditional parametric TPP models, where finding an efficient posterior sampling via MCMC for the latent dimensions is one main line of research [17]. Yet, to the best of our knowledge, there doesn't exist a computationally efficient Cox processes implementation. For example, the training times reported for a two-layer model in [17] are about 1-2 orders of magnitude higher than any of our applied models, ultimately making a hyperparameter search and comparison across 13 benchmark datasets infeasible. Please let us know if there is any specific efficient implementation we can compare with.
>
> > Q.E $n$
>
> We report this hyperparameter in Sec. 5 and use $n=100$ for all experiments.
>
> ### References
> 1. Ghatak, Ajoy, et al. "The dirac delta function." Quantum Mechanics: Theory and Applications (2004): 3-18.
> 2. Ho et al., [16] in the paper
> 3. Daley and Vere-Jones, [6] in the paper

---

> > ### Comment · Reviewer_1d9q · 2023-08-14
> > **Thank you for your response.**
> >
> > Thank you for your clarification. I found it helpful and clear. Therefore, I keep my recommendation and will stay tuned about the discussion from other reviewers on your interesting work.

---

### Author Rebuttal · Authors · 2023-08-09

We thank the reviewers for their valuable feedback and appreciation of the contribution and originality of our novel diffusion-based TPP model. We have attached a PDF with the model's sampling runtime and would further like to highlight parts of our answers to reviewer 1d9q (ELBO), reviewer 1xQ2 (Design choices), and reviewer RwFj (Model and method description).

In the following, we re-introduce a high-level explanation of the reverse process and the treatment of sets A-F that we will include in Sec. 3.2 in the updated version of the paper. We want to emphasize that the following neither introduces new notations nor new findings and solely presents Sec. 3.2 to be more accessible to the reader.

**Reverse process and case distinction**

To sample realizations $\bf{t}\sim\lambda_0$ starting from $\mathbf{t}^{(N)}\sim\lambda_{HPP}$, we need to learn to reverse the Markov chain of the forward process. Conditioned on $\mathbf{t}^{(0)}$, the reverse process at step $n$ is given by the posterior $q(\mathbf{t}^{(n-1)}|\mathbf{t}^{(0)},\mathbf{t}^{(n)})$, which is an inhomogeneous Poisson process for the chosen forward process. Therefore, the posterior can be represented by a history-independent intensity function $\lambda_{n-1}(t|\mathbf{t}^{(0)},\mathbf{t}^{(n)})$. As the forward process is defined by adding and thinning event sequences, the posterior over the random sequence $\mathbf{t}^{(n-1)}$ can be decomposed into disjoint sets of points based on whether they are also in $\mathbf{t}^{(0)}$ or $\mathbf{t}^{(n)}$. We use this case distinction to derive the posterior intensity and distinguish the following cases: points in $\mathbf{t}^{(n-1)}$ that where kept from $0$ to $n$ (**B**), points in $\mathbf{t}^{(n-1)}$, that were kept from $0$ to $n-1$ but thinned at the $n$-th step (**C**), added points in $\mathbf{t}^{(n-1)}$ that are kept in the $n$-th step (**D**) and added points in $\mathbf{t}^{(n-1)}$ that are thinned in the $n$-th step (**E**). Since the sets **B-E** are disjoint, the posterior intensity is a superposition of the intensities of each subset of $\mathbf{t}^{(n-1)}$: $\lambda_{n-1}(t|\mathbf{t}^{(0)},\mathbf{t}^{(n)})=\lambda^{(B)}(t)+\lambda^{(C)}(t)+\lambda^{(D)}(t)+\lambda^{(E)}(t)$.

To derive the intensity functions for cases B-E, we additionally define the following helper sets: **A** the points $\mathbf{t}^{(0)}\setminus\mathbf{t}^{(n-1)}$ that were thinned until $n-1$ and **F** the points $\mathbf{t}^{(n)}\setminus\mathbf{t}^{(n-1)}$ that have been added at step $n$. The full case distinction is further illustrated in Fig. 2. In the following paragraphs, we derive the intensity functions for cases B-E:

**B**: The set $\mathbf{t}^{(B)}$ can be formally defined as $\mathbf{t}^{(0)}\cap\mathbf{t}^{(n)}$ as $(\mathbf{t}^{(0)}\cap\mathbf{t}^{(n)})\setminus\mathbf{t}^{(n-1)}=\emptyset$, almost surely. This is because adding points at any of the locations $t\in\mathbf{t}^{(0)}\cap\mathbf{t}^{(n)}$ carries zero measure at every noising step. Hence, given $\mathbf{t}^{(0)}\cap\mathbf{t}^{(n)}$ the intensity can be written as a sum of Dirac measures: $\lambda^{(B)}(t)=\sum_{t_i\in(\mathbf{t}^{(0)}\cap\mathbf{t}^{(n)})}\delta_{t_i}(t)$.

**C**: Given $\mathbf{t}^{(A\cup C)}=\mathbf{t}^{(0)}\setminus\mathbf{t}^{(n)}$, $\mathbf{t}^{(C)}$ can be found by thinning and consists of points that were kept by step $n-1$ and removed at step $n$. Using the thinning of Eq. 2 \& 3, we know the probability of a point from $\mathbf{t}^{(0)}$ being in $\mathbf{t}^{(C)}$ and $\mathbf{t}^{(A\cup C)}$ is $\bar\alpha_{n-1}(1-\alpha_n)$ and $1-\bar\alpha_{n}$, respectively. Since we already know $\mathbf{t}^{(B)}$ we can consider the probability of finding a point in $\mathbf{t}^{(C)}$, given $\mathbf{t}^{(A\cup C)}$, which is equal to $\frac{\bar\alpha_{n-1}-\bar\alpha_{n}}{1-\bar\alpha_{n}}$.

**E**: The set $\mathbf{t}^{(E)}$ contains all points $t \notin (\mathbf{t}^{(0)}\cup \mathbf{t}^{(n)})$ that were added until step $n-1$ and thinned at step $n$. Again using Eq. 2 \& 3, we can see that these points were added with intensity $(1-\bar\alpha_{n-1})\lambda_{HPP}$ and then removed with probability $(1-\alpha_n)$ at the next step. Equivalently, we can write down the intensity that governs this process as $\lambda^{(E)}=(1-\bar\alpha_{n-1})(1-\alpha_n)\lambda_{HPP}$, i.e., sample points from an HPP and thin them to generate a sample $\mathbf{t}^{(E)}$.

**D**: The set $\mathbf{t}^{(D)}$ can be found by thinning $\mathbf{t}^{(D\cup F)}=\mathbf{t}^{(n)}\setminus\mathbf{t}^{(0)}$ and contains the points that were added by step $n-1$ and then kept at step $n$. The processes that generated $\mathbf{t}^{(D)}$ and $\mathbf{t}^{(F)}$ are two independent HPPs with intensities $\lambda^{(D)}=(1-\bar\alpha_{n-1})\alpha_n\lambda_{HPP}$ and $\lambda^{(F)}=(1-\alpha_{n})\lambda_{HPP}$, respectively, where $\lambda^{(D)}$ is derived similar to $\lambda^{(E)}$. Since $\mathbf{t}^{(D)}$ and $\mathbf{t}^{(F)}$ are independent HPPs and we know $\mathbf{t}^{(D\cup F)}$, the number of points in $\mathbf{t}^{(D)}$ follows a Binomial distribution with probability $p=\frac{\lambda^{(D)}}{\lambda^{(D)}+\lambda^{(F)}}$ (see A.2 for details). That means we can sample $\mathbf{t}^{(D)}$ given $\mathbf{t}^{(n)}$ and $\mathbf{t}^{(0)}$ by simply thinning $\mathbf{t}^{(D\cup F)}$ with probability $1-p$ and express the intensity as a thinned sum of Dirac measures (c.f., Fig. 2).

For sequences of the training set, where $\mathbf{t}^{(0)}$ is known, we can compute these intensities for all samples $\mathbf{t}^{(n)}$ and reverse the forward process. However, $\mathbf{t}^{(0)}$ is unknown when sampling new sequences. Therefore, similarly to the denoising diffusion approaches [16], in the next section, we show how to approximate the posterior intensity, given only $\mathbf{t}^{(n)}$. Further, in Sec. 3.4, we demonstrate how the trained neural network can be leveraged to sample new sequences $\mathbf{t}\sim\lambda_0$.

---

### Decision · Program_Chairs · 2023-09-21

**Decision:**

Accept (poster)

**Comment:**

The paper presents a novel idea - a diffusion model for temporal point process. The original paper had clarity issues, but the authors responded well to reviewers' comments and now the scores seem to have moved towards acceptance. Still some concerns about clarity and presentation remain. I recommend accept as a poster based on the novelty.